# LEARNING MORE SKILLS THROUGH OPTIMISTIC EXPLORATION

**DJ Strouse**,[*] **Kate Baumli, David Warde-Farley, Vlad Mnih, Steven Hansen**[*]
DeepMind
`{strouse, baumli, dwf, vmnih, stevenhansen}@google.com`

## ABSTRACT

Unsupervised skill learning objectives (Eysenbach et al., 2019; Gregor et al., 2016) allow agents to learn rich repertoires of behavior in the absence of extrinsic rewards. They work by simultaneously training a policy to produce distinguishable latent-conditioned trajectories, and a discriminator to evaluate distinguishability by trying to infer latents from trajectories. The hope is for the agent to explore and master the environment by encouraging each skill (latent) to reliably reach different states. However, an inherent exploration problem lingers: when a novel state is actually encountered, the discriminator will necessarily not have seen enough training data to produce accurate and confident skill classifications, leading to low intrinsic reward for the agent and effective penalization of the sort of exploration needed to actually maximize the objective. To combat this inherent pessimism towards exploration, we derive an information gain auxiliary objective that involves training an ensemble of discriminators and rewarding the policy for their disagreement. Our objective directly estimates the epistemic uncertainty that comes from the discriminator not having seen enough training examples, thus providing an intrinsic reward more tailored to the true objective compared to pseudocount-based methods (Burda et al., 2019). We call this exploration bonus **dis**criminator **disa**greement **in**trinsic reward, or **DISDAIN**. We demonstrate empirically that DISDAIN improves skill learning both in a tabular grid world (Four Rooms) and the 57 games of the Atari Suite (from pixels). Thus, we encourage researchers to treat pessimism with DISDAIN.

## 1 INTRODUCTION

Reinforcement learning (RL) has proven itself capable of learning useful skills when clear task-specific rewards are available (OpenAI et al., 2019a;b; Vinyals et al., 2019). However, truly intelligent agents should, like humans, be able to learn even in the absence of supervision in order to acquire repurposable task-agnostic knowledge. Such unsupervised pre-training has seen recent success in language (Radford et al., 2019; Brown et al., 2020) and vision (Chen et al., 2020a;b), but its potential has yet to be fully realized in the learning of behavior.

The most promising class of algorithms for unsupervised skill discovery is based upon maximizing the *discriminability* of skills represented by latent variables on which a policy is conditioned. Objectives are typically derived from variational approximations to the mutual information between latent variables and states visited (Gregor et al., 2016; Eysenbach et al., 2019; Warde-Farley et al., 2019; Hansen et al., 2020; Baumli et al., 2021), employing a learned parametric skill discriminator. Both the policy and discriminator have the objective of strong predictive performance by the discriminator, though the discriminator is trained with supervised learning rather than RL. The end result is a policy capable of producing consistently distinguishable behavioral motifs, or "skills." These skills can be evaluated zero-shot, fine-tuned, or composed in a hierarchical RL setup to maximize task reward when one is introduced (Eysenbach et al., 2019; Hansen et al., 2020). Unsupervised skill learning objectives can also be maximized *in conjunction with* task reward, in order to promote robustness of learned behavior to environment perturbations (Mahajan et al., 2019; Kumar et al., 2020).

---

[*]equal contribution

The degree to which unsupervised skill discovery methods are useful in such downstream applications depends on how many skills they are able to learn. Indeed, Eysenbach et al. (2019) showed that as more skills are learned, the performance obtained using the learned skills in a hierarchical reinforcement learning setup improves. In follow up work, Achiam et al. (2018) showed that methods like DIAYN can struggle to learn large numbers of skills and proposed gradually increasing the number of skills according to a curriculum to make skill discovery easier.

The aim of this work is to improve the ability of skill discovery methods to learn more skills. We highlight an exploration problem intrinsic to the entire class of variational skill discovery algorithms which can inhibit discovery of new skills. During skill learning, the policy will necessarily need to explore new states of the environment. The discriminator must then make latent predictions for states it has never seen before, resulting in incorrect and/or low-confidence predictions. The policy will in turn be penalized for this poor discriminator performance, and discouraged from seeking out new states. We refer to this problem as "pessimistic exploration" and describe it further in section 2.

To motivate our solution, we argue that it is important for the policy to distinguish between two kinds of uncertainty in the discriminator - *aleatoric* uncertainty that comes from the policy producing similar trajectories for different skills, and *epistemic* uncertainty that comes from a lack of training data. The former indicates poor policy performance, but the latter is in fact desirable, and serves as a signal for potential discriminator learning. Unfortunately, skill discovery algorithms treat both types of uncertainty the same and thus ignore this important signal. We propose to capture it.

Our primary contribution, which we present in section 3, is an exploration bonus tailored to skill discovery algorithms, designed to overcome pessimistic exploration. We identify states of high epistemic uncertainty in the discriminator by training an ensemble of discriminators and measuring their disagreement. We call this exploration bonus **dis**criminator **dis**agreement **in**trinsic reward, or **DISDAIN**. Intuitively, the ensemble members may disagree in novel states, but must come to agree in frequently visited ones. More formally, we derive DISDAIN from a Bayesian perspective that 1) represents the posterior over discriminator parameters using an ensemble, and 2) encourages the policy to maximize information gain (i.e. reduce uncertainty) about the parameters of the discriminator. In section 4, we demonstrate empirically that DISDAIN improves skill learning over unbonused skill discovery algorithms in both an illustrative grid world (Four Rooms, Sutton et al. (1999)) and the Atari 2600 learning environment (ALE, Bellemare et al. (2013)) more so than augmenting with popular exploration bonuses not tailored to skill discovery.

## 2 UNSUPERVISED SKILL LEARNING THROUGH VARIATIONAL INFOMAX

### 2.1 INTRODUCTION

We now formalize our setting of interest: unsupervised skill learning through variational information maximization (Gregor et al., 2016; Eysenbach et al., 2019; Warde-Farley et al., 2019; Hansen et al., 2020; Baumli et al., 2021). We consider a Markov decision process (MDP), defined by the tuple $\mathcal{M} = (\mathcal{S}, \mathcal{A}, p_E, \rho, r, \gamma)$, where $\mathcal{S}$ and $\mathcal{A}$ are state and action spaces, respectively, the environment dynamics $p_E(s' \mid s, a)$ specifies the probability of transitioning to state $s' \in \mathcal{S}$ when taking action $a \in \mathcal{A}$ in state $s \in \mathcal{S}$, $\rho(s)$ denotes the probability of starting an episode in state $s$, and $\gamma \in [0, 1)$ is a discount factor. Since we focus on unsupervised training, we ignore the environmental reward $r$.

Our agents seek to learn a repertoire of skills, indexed by the latent variable $Z$ and represented by the policy $\pi_\theta(a \mid s, z)$, which is parameterized by $\theta$ and maps from states and latent variables to distributions over actions. The latent variables are sampled $z \sim p(Z)$ at the beginning of each trajectory and then fixed, so each $z$ represents a temporally extended behavior. The skill trajectory length $T$ may differ from the episode length, thus a new skill might be resampled within an episode.

For conciseness, we will denote trajectories sampled from the policy by $\tau \sim \pi(z)$ when conditioning on a particular skill $z$, and $\tau \sim \pi$ when collecting trajectories across skills. To simplify our discussion, and because it is the most common case in practice, we will assume that $Z$ is categorical with cardinality $N_Z$, although much of the discussion carries over to continuous $Z$.

A large and growing class of objectives for unsupervised skill discovery are derived from maximizing the mutual information between the latent skill $Z$ and some feature of the resulting trajectories $O(\tau)$:

$$\mathcal{F}(\theta) \equiv I(Z, O) = H(Z) - H(Z \mid O) = \mathbb{E}_{z \sim p(z), \tau \sim \pi(z)}[\log p(z \mid o(\tau)) - \log p(z)]. \quad (1)$$

For example, variational intrinsic control (VIC, Gregor et al. (2016)) maximizes $I(Z; S_0, S_T)$ - the mutual information between the skill and initial and final states ($o_T = (s_0, s_T)$).[1] Diversity is all you need (DIAYN, Eysenbach et al. (2019)), on the other hand, maximizes $I(Z, S)$ - the mutual information between the skill and each state along the trajectory ($o_t = s_t$ for $t \in 1 : T$). Intuitively, VIC produces skills that vary in the destination reached (without regard for the path taken), while DIAYN produces skills that vary in the path taken (with less emphasis on the destination reached).

In practice, maximizing equation 1 is not straightforward, because it requires calculating the conditional distribution $p(Z \mid O)$. In general, it is necessary to estimate it with a learned parametric model $q_\phi(Z \mid O)$. We refer to this model as the *discriminator*, since it is trained to discriminate between skills. Fortunately, replacing $p$ with $q$ still yields a lower bound on $\mathcal{F}(\theta)$ (Barber and Agakov, 2004), and we may instead maximize the proxy objective $\tilde{\mathcal{F}}(\theta)$:

$$\mathcal{F}(\theta) \geq \tilde{\mathcal{F}}(\theta) = \mathbb{E}_{z \sim p(z), \tau \sim \pi(z)}[\log q_\phi(z \mid o(\tau)) - \log p(z)]. \tag{2}$$

Optimizing $\tilde{\mathcal{F}}(\theta)$ with respect to the policy parameters $\theta$ corresponds to RL on the reward:

$$r_{\text{skill}} = \log q_\phi(z \mid o) - \log p(z). \tag{3}$$

Since the intent is for the agent to learn a full repertoire of skills, the skill prior $p(z)$ is typically fixed to be uniform (Eysenbach et al., 2019; Achiam et al., 2018; Baumli et al., 2021), in which case $-\log p(z) = \log N_Z$. If the discriminator simply ignores the trajectory and guesses skills uniformly as well, then $\log q_\phi(z \mid o) = -\log N_Z$ and the reward will be zero. If the discriminator instead guesses perfectly, then $\log q_\phi(z \mid o) = 0$, and the reward will be $\log N_Z$. More generally, the expected reward is an estimate of the logarithm of the *effective* number of skills. Thus, measuring the reward in bits (i.e. using $\log_2$ in equation 3), we can estimate the number of skills learnt as:

$$n_{\text{skills}} = 2^{\mathbb{E}[r_{\text{skill}}]}. \tag{4}$$

We adopt this quantity as our primary performance metric for our experiments.

To make sure the bound in equation 2 is as tight as possible, the discriminator $q_\phi(Z \mid O)$ must also be fit to its target $p(Z \mid O)$ through supervised learning (SL) on the negative log likelihood loss:

$$L(\phi) \equiv -\mathbb{E}_{z \sim p(z), \tau \sim \pi(z)}[\log q_\phi(z \mid o(\tau))]. \tag{5}$$

The loss is minimized, and the bound in equation 2 tight, when $q_\phi(Z \mid O) = p(Z \mid O)$.

The joint optimization of $\tilde{\mathcal{F}}(\theta)$ by RL and $L(\phi)$ by SL forms a cooperative communication game between policy and discriminator. The agent samples a skill $z \sim p(Z)$ and generates the "message" $\tau \sim \pi(z)$. The discriminator receives the message $\tau$ and attempts to decode the original skill $z$. When the policy produces trajectories for different skills that do not overlap in the features $O(\tau)$, the discriminator will easily learn to label trajectories, and when the discriminator makes accurate and confident predictions, the reward in equation 3 will be high. Ideally, the end result is a policy exhibiting a maximally diverse set of skills. This joint-training system is depicted in Figure 2a.

## 2.2 PESSIMISTIC EXPLORATION IN UNSUPERVISED SKILL DISCOVERY

A conflict arises between the exploration necessary for skill diversification and rewards supplied by an imperfect discriminator, trained only on past policy-generated experience. Without data (and in the absence of perfect generalization), the discriminator is likely to make poor predictions when presented with trajectories containing previously unseen states, resulting in low reward for the policy. Importantly, this penalization occurs *regardless* of whether the policy produces distinguishable skill-conditioned trajectories if the current discriminator is a locally poor approximation to $p(Z \mid O)$ for a region of state space represented in $O$. We note that this is distinct from issues of pessimism in exploration that arise more generally, including with stationary reward functions (Osband et al., 2019), wherein naive exploration strategies fail to adequately position an agent for further acquisition of information. In the scenario we examine here, the agent's sole source of supervision directly sabotages the learning process when the discriminator extrapolates poorly.

---

[1] More accurately, Gregor et al. (2016) *conditioned* on initial state and used $I(Z, S_T \mid S_0) = H(Z \mid S_0) - H(Z \mid S_0, S_T)$. However, it has subsequently become common not to condition the skill sampling distribution on $s_0$ (Eysenbach et al., 2019), in which case $H(Z \mid S_0) = H(Z)$ and $I(Z, S_T \mid S_0) = I(Z; S_0, S_T)$.

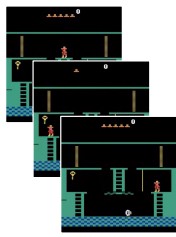

$\tau \sim \pi(z_1)$    $\tau \sim \pi(z_2)$    $\tau \sim \pi(z_1)$

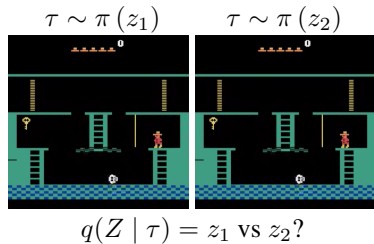

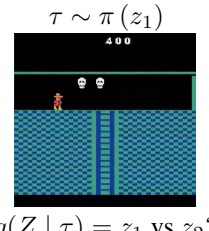

$q(Z \mid \tau) = z_1$ vs $z_2$?    $q(Z \mid \tau) = z_1$ vs $z_2$?

(a) *Past trajectories.*     (b) *Aleatoric uncertainty* arises when different skills visit similar states.     (c) *Epistemic uncertainty* arises from exploring novel states.

Figure 1: **The pessimistic exploration problem**. Because skill discovery objectives do not distinguish between aleatoric and epistemic uncertainty, they penalize exploration.

We argue that in order to overcome this inherent pessimism, we must distinguish between two kinds of uncertainty in the discriminator: *aleatoric* uncertainty (figure 1b) that is due to the policy producing overlapping skills (i.e. high $H(Z \mid O)$), and *epistemic* uncertainty (figure 1c) that is due to a lack of training data (i.e. poor match between $q_\phi(Z \mid O)$ and $p(Z \mid O)$). Naively, both kinds of uncertainty contribute to low reward for the policy (equation 3), but while reduction of aleatoric uncertainty requires changes by the policy, epistemic uncertainty can be reduced (and thus reward increased) simply by creating more data (i.e. visiting the same states again). Thus, we argue that we should "reimburse" the policy for *epistemic* uncertainty in the discriminator, and in fact *encourage* the policy to visits states of high epistemic uncertainty.

Put another way, when the policy only maximizes the reward of equation 3, it maximizes the lower bound $\tilde{\mathcal{F}}(\theta) \leq \mathcal{F}(\theta)$ without regard for its tightness, despite that a looser bound means a more pessimistic reward. The job of keeping the bound tight is left entirely to the discriminator (equation 5). By incentivizing the policy to visit states of high epistemic uncertainty, valuable training data is provided to the discriminator that allows it to better approximate its target and close the gap between $\tilde{\mathcal{F}}(\theta)$ and $\mathcal{F}(\theta)$. In this way, we encourage the policy to help keep the bound $\tilde{\mathcal{F}}(\theta) \leq \mathcal{F}(\theta)$ tight.

In the next section, we formalize the intuitions outlined in the last two paragraphs. The result is an exploration bonus measured as the disagreement among an ensemble of discriminators.

## 3    DISDAIN: DISCRIMINATOR DISAGREEMENT INTRINSIC REWARD

To formalize the notion of discriminator uncertainty, we take a Bayesian approach and replace the point estimate of discriminator parameters $\phi$ with a posterior $p(\phi)$. Maintaining a posterior allows us to quantify the information gained about those parameters. Specifically, we will incentivize the policy to produce trajectories in which observing the paired skill label $z$ provides maximal information about the discriminator parameters $\phi$:

$$I(Z; \Phi \mid O) = H(Z \mid O) - H(Z \mid O, \Phi). \tag{6}$$

Rewriting the entropies as expectations over trajectories, we have:

$$I(Z; \Phi \mid O) = \mathbb{E}_{\tau \sim \pi}\left[ H\left[ \int p(\phi)\, q_\phi(Z \mid o(\tau))\, d\phi \right] - \int p(\phi)\, H[q_\phi(Z \mid o(\tau))]\, d\phi \right] \tag{7}$$

where we have used that $p(\phi)$ does not depend on the present trajectory and so $p(\phi \mid o(\tau)) = p(\phi)$. Note that unlike in equations 2 and 5, the expectation over trajectories is not skill-conditioned. The marginalization over $Z$ happens in the entropy and is over the *discriminator's posterior* $q_\phi(z \mid s)$ rather than the agent's prior $p(z)$. This is because the discriminator parameters $\Phi$ are now part of the probabilistic model and do not just enter through a variational approximation.

How should we represent the posterior over discriminator parameters $p(\phi)$? There is considerable work in Bayesian deep learning that offers possible answers, but here we take an ensemble approach (Seung et al., 1992; Lakshminarayanan et al., 2017). We train $N$ discriminators with parameters $\phi_i$ for the $i^{\text{th}}$ discriminator. The discriminators are independently initialized and in theory could also be trained on different mini-batches, though in practice, we found it both sufficient and

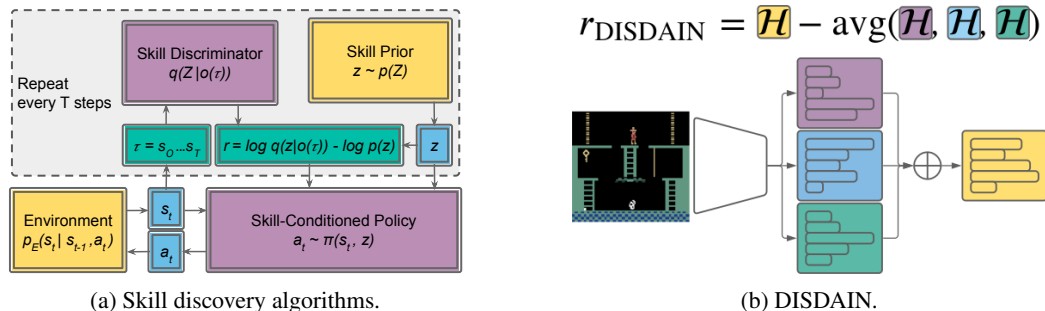

(a) Skill discovery algorithms.              (b) DISDAIN.

Figure 2: **Methods**. (a) The skill discovery process, where joint optimization of a skill-conditioned policy and skill discriminator ensure reliable and distinct behavior for each skill. (b) DISDAIN: disagreement between an ensemble of skill discriminators informs exploration.

simpler to train them on the same mini-batches, as others have also found (Osband et al., 2016). The posterior $p(\phi)$ is then represented as a mixture of point masses at the $\phi_i$:

$$p(\phi) = \frac{1}{N} \sum_{i=1}^{N} \delta(\phi - \phi_i). \tag{8}$$

Substituting the posterior in equation 8 into equation 7, we have:

$$I(Z; \Phi \mid O) = \mathbb{E}_{\tau \sim \pi} \left[ H\left[ \frac{1}{N} \sum_{i=1}^{N} q_{\phi_i}(Z \mid o(\tau)) \right] - \frac{1}{N} \sum_{i=1}^{N} H[q_{\phi_i}(Z \mid o(\tau))] \right]. \tag{9}$$

Maximizing equation 9 with RL corresponds to adding the following auxiliary reward to the policy:

$$r_{\text{DISDAIN}}(t) \equiv H\left[ \frac{1}{N} \sum_{i=1}^{N} q_{\phi_i}(Z \mid o_t) \right] - \frac{1}{N} \sum_{i=1}^{N} H[q_{\phi_i}(Z \mid o_t)]. \tag{10}$$

In words, this is the *entropy of the mean* discriminator minus the *mean of the entropies* of the discriminators. By Jensen's inequality, entropy increases under averaging, and thus $r_{\text{DISDAIN}} \geq 0$.

For trajectories on which there has been ample training data for the discriminators, the ensemble members should agree and $q_{\phi_i}(z \mid o) \approx q_{\phi_j}(z \mid o)$ for all $i, j$. Thus the two terms in equation 10 will be equal and this reward will vanish. For states of high *discriminator disagreement*, however, this reward will be positive, encouraging exploration. Therefore, we call equation 10 the Discriminator Disagreement Intrinsic reward, or DISDAIN (figure 2b).

DISDAIN is simple to calculate for discrete $Z$, as is common in the relevant literature (Gregor et al., 2016; Eysenbach et al., 2019; Achiam et al., 2018; Baumli et al., 2021). DISDAIN augments any discriminator-based unsupervised skill learning algorithm with two changes. First, an ensemble of discriminators is trained instead of just one, and $r_{\text{skill}}$ should be calculated using the ensemble-averaged prediction $q_\phi(Z \mid O) = \frac{1}{N} \sum_{i=1}^{N} q_{\phi_i}(Z \mid O)$. Second, the DISDAIN reward is combined with $r_{\text{skill}}$, which we do through simple addition with a tunable multiplier $\lambda$. Pseudocode for DISDAIN is provided in Algorithm 1. With $N = 1$ and $\lambda = 0$, this is standard unbonused skill discovery.

## 4 EXPERIMENTS

We validate DISDAIN by testing its ability to increase skill learning in an illustrative grid world (Four Rooms) as well as a more challenging pixel-based setting requiring function approximation (the 57 Atari games of the Arcade Learning Environment (Bellemare et al., 2013)). In addition to comparing performance to unbonused skill learning, we also compare to using popular off-the-shelf exploration bonuses that are not tailored to skill discovery. In Four Rooms, we compare to using count-based bonuses, which are known to perform well in these settings (Brafman and Tennenholtz, 2002). In Atari, where count-based bonuses become untenable due to the enormous state space, we compare to random network distillation (RND; Burda et al. (2019)), one of the most commonly used

---

**Algorithm 1:** Skill discovery with DISDAIN

---

**Input:** policy $\pi_\theta$, discriminator ensemble $\{q_{\phi_i}\}_{i=1}^N$, skill features $O(\tau)$, skill distribution $p(Z)$, skill trajectory length $T$, DISDAIN reward weight $\lambda$

**while** *not converged* **do**
  Reset environment, sampling initial state $s_0$
  **while** *episode not ended* **do**
    Sample skill, $z \sim p(Z)$
    Sample trajectory of length $T$ from $s_0$, $\tau \sim \pi(z)$
    Form average discriminator from ensemble, $q_\phi = \frac{1}{N} \sum_{i=1}^N q_{\phi_i}$
    $r_{\text{skill}} = \log q_\phi(z \mid O(\tau)) - \log p(z)$
    $r_{\text{DISDAIN}} = H[q_\phi(\cdot \mid O(\tau))] - \frac{1}{N} \sum_{i=1}^N H[q_{\phi_i}(\cdot \mid O(\tau))]$
    $r = r_{\text{skill}} + \lambda r_{\text{DISDAIN}}$
    Update $\theta$ with RL to maximize $r$
    Update $\{q_{\phi_i}\}_{i=1}^N$ with SL to maximize $\log q_{\phi_i}(z \mid O(\tau))$
    $s_0 = s_T$

---

pseudo-count based methods (Bellemare et al., 2016). In addition, we validate that any advantages of DISDAIN are not purely due to using an ensemble of discriminators by evaluating an ablation that includes the same ensemble but removes the DISDAIN reward (i.e. Algorithm 1 with $\lambda = 0$). In all cases, our primary metric for comparison is the effective number of skills learnt, $n_{\text{skills}}$ (equation 4).

The effective number of skills is just an interpretable transformation of the mutual information objective shared by a large number of unsupervised skill learning algorithms (e.g. Gregor et al. (2016); Achiam et al. (2018); Eysenbach et al. (2019); Hansen et al. (2020)). As such, the fact that DISDAIN helps better maximize this objective should be worthwhile in its own right, considering the various use cases that motivated this objective in the existing literature. That said, we recognize that it is not obvious what utility comes with increasing the number of effective skills. To address this, we also measure three surrogates of skill utility: downstream goal achievement, unsupervised reward attainment, and unsupervised state coverage.

**Distributed training** We use a distributed actor-learner setup similar to R2D2 (Kapturowski et al., 2019), except we do not use replay prioritization or burn-in, and Q-value targets are computed with Peng's $Q(\lambda)$ (Peng and Williams, 1994) rather than $n$-step double Q-learning. In all cases, we found it important to train separate Q-functions for the skill learning rewards and exploration bonuses (DISDAIN, RND, or count-based). This follows from Burda et al. (2019), who also found that this setup helped stabilize learning. Unlike that work, we need to specify a value target for each Q-function, which we take to be the value of the action that maximizes the *composite* value function, where the two values are added with a tunable weight $\lambda$. The discriminator is trained on the same mini-batches sampled from replay to train the Q-functions. For Atari experiments, the Q-networks process batches of state, skill, and action tuples to produce scalar Q-values for each, and the ResNet state embedding network used in Espeholt et al. (2018) is shared by both of the Q-networks and discriminator. For the Four Rooms grid world, we use tabular Q-functions and discriminators. Further implementation details can be found in appendix A.

**Skill discovery** Our skill discovery baseline deviates slightly from prior work in order to make it representative of the skill discovery literature as a whole. We utilize a simple discriminator that receives only the final state of a trajectory (so $O(\tau) = s_T$), omitting the state-conditional prior and action entropy bonuses used in specific algorithms (Gregor et al., 2016; Eysenbach et al., 2019).

**Hyperparameters** Most of the RL hyperparameters were not tuned, but rather taken from standard values known to be reasonable for Peng's $Q(\lambda)$. The skill discovery specific hyperparameters were tuned for the basic algorithm without exploration bonuses, and then reused in all conditions. Our RND implementation was first tuned without skill learning to achieve Atari performance competitive with the original paper. For all exploration bonuses (DISDAIN, RND, count), we tuned their reward weighting (e.g. $\lambda$ in algorithm 1), while for DISDAIN we additionally swept the ensemble size ($N$).

**Baselines** As with the skill discovery reward, we apply exploration bonuses only to the terminal states of each skill trajectory. For the count-based bonus, we track the number of times an agent ends

a skill trajectory in each state $n(s)$ and apply the exploration bonus $r_T = 1/\sqrt{n(s_T)}$. For RND, we follow the details of Burda et al. (2019) as closely as possible. For our target and predictor networks, we use the same ResNet architecture as the policy observation embedding described above, and then project to a latent dimension of 128. Rather than normalizing observations based on running statistics, we found it more reliable to use the standard $\frac{1}{255}$ normalization of Atari observations.

**DISDAIN**   Our ensemble-based uncertainty estimator required many design choices, including the size of ensembles, and to what extent the ensemble members shared training data and parameters. In all of the domains we tested, we found training ensemble members on different batches to be unnecessary, similar to Osband et al. (2016). In the tabular case (Four Rooms), parameter-sharing is not a concern, and we found an ensemble size of $N = 2$ to be sufficient. For Atari, the ensemble of discriminators reuse the single ResNet state embedding network which is shared by the value function. The input to the ensemble will inevitably drift, even if the data distribution remains constant, since the ResNet representations evolve according to a combination of the value function and discriminator updates. Expressive ensembles (e.g. with hidden layers) never converged in practice. By contrast, large linear ensembles ($N = 40$) were reliably convergent, with convergence time increasing with ensemble size. We follow Osband et al. (2016) in scaling the gradients passed backward through the embedding network by $\frac{1}{40}$ to account for the increase in effective learning rate.

## 4.1   FOUR ROOMS

First, we evaluate skill learning in the illustrative grid world seen in figure 3. There are 4 rooms and 104 states. The agent begins each episode in the top left corner and at each step chooses an action from the set: `left`, `right`, `up`, `down`, or `no-op`. Episodes are 20 steps and we sample one skill per episode (i.e. $T = 20$). The episodes are long enough to reach all but one state, allowing for a maximum of 103 skills. For each method, we set $N_Z = 128$ to make this theoretically possible.[2]

**Results**   As seen in figure 3, even in this simple task, unbonused agents are unable to exceed 30 skills and barely leave the first room (see figures 18 and 19 for example skill rollouts). With both DISDAIN and a count bonus, agents explore all four rooms and learn approximately triple the number of skills, with the best seeds learning approximately 90 skills. Both bonuses do slow learning and add variance due to the addition of a separate learned Q-function (see figure 20 for individual seeds). The ensemble-only ablation provides a small benefit to skill learning, but far less than DISDAIN, demonstrating that the DISDAIN exploration bonus ($r_{\text{DISDAIN}}$), rather than the ensembling, drives the increased performance. As a simple demonstration of the usefulness of our learnt skills for downstream tasks, we also evaluate each methods' skills on the original Four Rooms reward (i.e. reaching specified goal states) without additional finetuning. Specifically, for each method, we sample each accessible state of the environment as a goal, pass it through the trained discriminator $q_\phi$, choose the highest probability skill $z$, rollout the policy for one episode conditioned on $z$, and track the fraction of goal states successfully reached (similar to the imitation learning evaluation of Eysenbach et al. (2019)). As seen in Figure 3b, more learnt skills leads to better downstream task performance.

## 4.2   ATARI

Next we consider skill learning on a standard suite of 57 Atari games, where prior work has shown that learning discrete skills is quite difficult (Hansen et al., 2020). In addition, it is a non-trivial test for our ensemble uncertainty based method to work in the function approximation setting, since this depends on how each ensemble member generalizes to unseen data. Here we use $N_Z = 64$ and $T = 20$. Since Atari episodes vary in length, skills may be resampled within an episode.

The count-based baseline used in the Four Rooms experiments cannot be directly applied to a non-tabular environment like Atari. While pseudo-count methods have been used here (Bellemare et al., 2016), Random Network Distillation (RND) similarly induces long term exploration (Burda et al., 2019) and has been used for this purpose in a state-of-the-art Atari agent (Badia et al., 2020). While newer approaches surpass RND in some domains (Raileanu and Rocktäschel, 2020; Seo et al., 2021), it is unclear if this is the case across the full Atari suite. So, at present, we believe RND is the most compelling baseline to compare against DISDAIN.

---

[2]An open source reimplementation of DISDAIN on a smaller version of Four Rooms is available at http://github.com/deepmind/disdain.

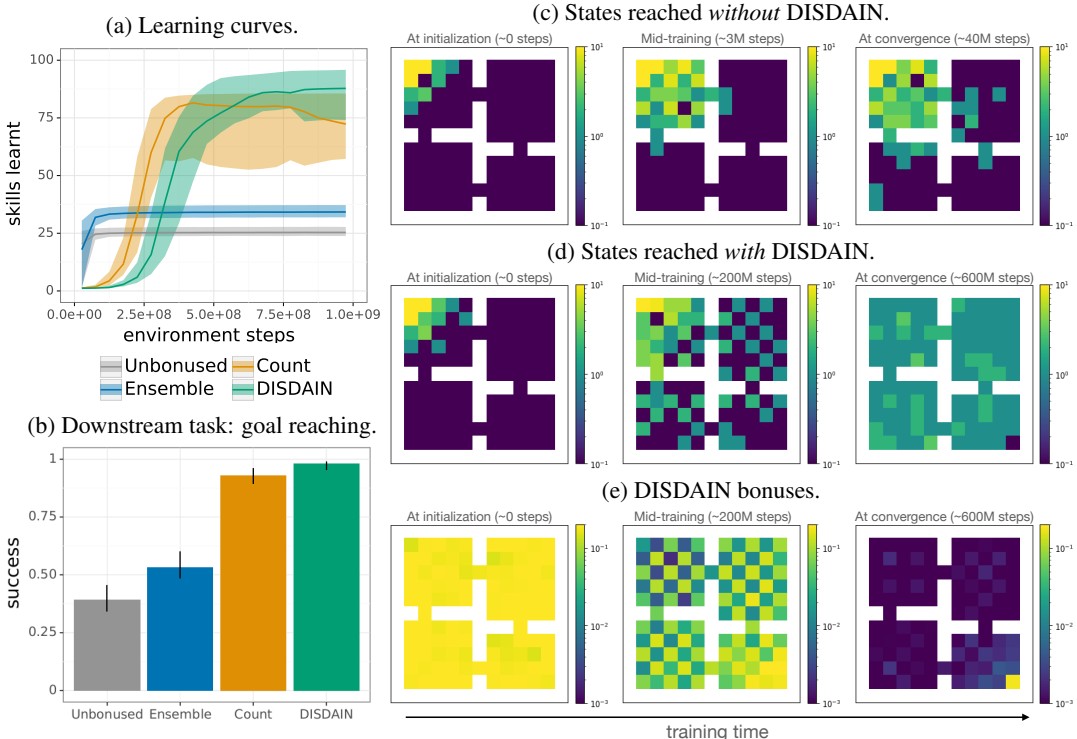

Figure 3: **Four Rooms results.** (a) Skills learnt for top 10 of 20 seeds for each method. Mean ± std over seeds. (b) Performance on the downstream task of reaching a target state, averaged across all possible target states. Mean ± std over seeds. (c-d) Example states reached with and without DISDAIN. Plots depict counts of final states reached after one rollout per skill. Columns correspond to different points during training. With DISDAIN, agents learn to reach all states, while without, they barely make it out of the first room. (e) Per-state DISDAIN bonuses for the policy depicted in (d). In the beginning, all exploration is encouraged. In the middle, the checkerboard pattern emerges because agents try to space out their skills. By the end of training, DISDAIN gracefully fades away.

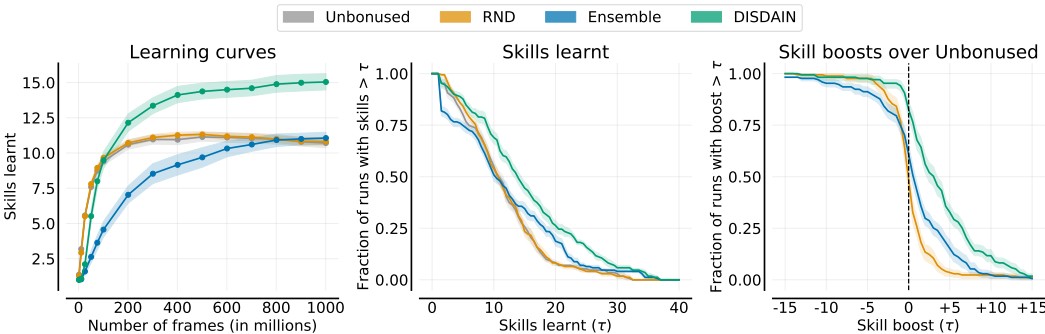

Figure 4: **Number of skills learnt on Atari**. *Left*: Effective skills (see equation 4) over training, measured by interquartile mean (IQM). *Center*: Distribution of skills learnt across seeds and games. *Right*: Distribution of skill boosts over Unbonused skill learning across tasks and games. Shaded regions show pointwise 95% confidence bands based on percentile bootstrap with stratified sampling.

**Results**    As pointed out in Agarwal et al. (2021), making statistically sound conclusions can be challenging in the few-seed, many-task setting of Atari. Thus, we follow their recommendations and focus our results primarily on statistically robust distributional claims here. Individual learning curves and game-by-game results are available in the appendix.

As shown in figure 4, DISDAIN increases the effective number of skills learnt across the Atari suite, with only modest damage to sample effiency. Additionally, we found that DISDAIN's performance

boosts are robust to its key hyperparameters, namely the bonus weight $\lambda$ and ensemble size $N$ (see figure 8), with significant gains over unbonused skill learning maintained over more than an order of magnitude of variation in both.

Notably, our results show that RND fails to significantly aid in skill learning. A sweep over the RND bonus weight is shown in figure 11, but the summary is that as the RND bonus becomes similar in magnitude to the skill learning reward, it damages skill learning rather than helping, and so the "best" RND bonus weight for skill learning is approximately zero. This is perhaps unsurprising: RND was designed in the context of stationary task rewards (as are most other exploration methods, such as pseudo-counts), whereas skill learning objectives produce a highly non-stationary reward function. This highlights the importance of using an exploration bonus tailored to the skill discovery setting. Additionally, the failure of the "ensemble-only" baseline to significantly increase skill learning demonstrates that is the $r_{\text{DISDAIN}}$ exploration bonus that is crucial to DISDAIN's success, and not just the ensembling of the discriminator.

To further probe the utility of our learnt skills, we measure the unsupervised reward attainment of a policy that randomly switches between them. First reported in Hansen et al. (2020), the idea is that while this policy is certainly far from optimal, this metric indicates whether or not the skill space is sufficient to perform the various reward collecting behaviors involved in each game. Additionally, we measure lifetime state coverage as an indication of exploration throughout learning (on a subset of games supporting this metric; see Appendix B for details). Figure 5 confirms that DISDAIN leads to both increased reward attainment and lifetime coverage (see figures 14, 15, and 16 for additional lifetime and episodic state coverage results).

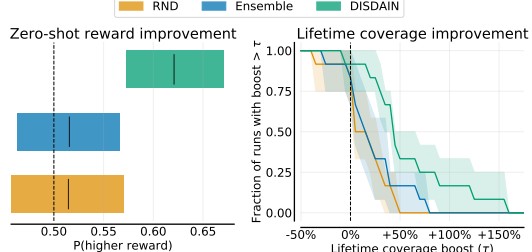

Figure 5: **Qualitative analysis of skills learnt on Atari**. Both plots depict improvement over Unbonused skill learning aggregated across seeds and games. *Left*: Probability of improvement on zero-shot reward evaluation. *Right*: Distribution of percentile improvements on lifetime coverage. Error bars depict 95% stratified boostrap CIs.

## 5 DISCUSSION & LIMITATIONS

We introduced DISDAIN, an enhancement for unsupervised skill discovery algorithms which increases the effective number of skills learnt across a diverse range of tasks, by using ensemble-based uncertainty estimation to counteract a bias towards pessimistic exploration.

The connection between ensemble estimates of uncertainty and infomax exploration dates back to at least Seung et al. (1992), who use it to select examples in an active supervised learning setting. These ideas have more recently found use in the RL literature, with recent work using disagreement among ensembles of value functions (Chen et al., 2017; Flennerhag et al., 2020; Zhang et al., 2020) and forward models of the environment (Pathak et al., 2019; Shyam et al., 2019; Sekar et al., 2020) to drive exploration. In a closely related line of work, ensembles of Q-functions have been used to drive exploration without an explicit bonus based on disagreement (Osband et al., 2016; 2018). To our knowledge, DISDAIN represents the first application of these ideas to unsupervised skill discovery.

We focused on discrete skill learning methods due to their relative prevalence (Gregor et al., 2016; Eysenbach et al., 2019; Achiam et al., 2018; Baumli et al., 2021). In some cases, continuous or structured skill spaces might make more sense (Hansen et al., 2020; Warde-Farley et al., 2019). While the principles behind DISDAIN should still apply, further approximations may be necessary, e.g. for the entropy of the ensemble-averaged discriminator, which may be unavailable in closed form.

Designing agents that explore and master their environment in the absence of task reward remains an open problem, for which there exist many different families of approaches (e.g. reward-free exploration (Jin et al., 2020; Zhang et al., 2021)). In this paper, we focus on improving one such family - unsupervised skill learning through variational infomax (Section 2). By treating agents with DISDAIN, we empower them to better maximize their objective and learn more skills. Leveraging these skills for rapid task reward maximization remains an important direction for future research.

## ACKNOWLEDGEMENTS

The authors would like to thank Stephen Spencer for engineering and technical support, Ian Osband for feedback on an early draft, Rishabh Agarwal for suggestions on statistical analysis and plotting, Phil Bachman for correcting the error discussed in Appendix C, and David Schwab for pointing us to the original Query by Committee work (Seung et al., 1992).

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

## A  COMPUTE REQUIREMENTS AND HYPERPARAMETERS

The compute cluster we performed experiments on is rather heterogeneous, and has features such as host-sharing, adaptive load-balancing, etc. It is therefore hard to give precise details regarding compute resources – however, the following is a best-guess estimate.

A full experimental training run for Atari lasted 4 days on average. Our distributed reinforcement learning setup (Espeholt et al., 2018) used 100 CPU actors and a single V100 GPU learner. Thus, we required approximately 9600 CPU hours and 96 V100 GPU hours per seed, with 3 seeds and 3 conditions per game.

Tuning required approximately 10 different hyper-parameters combinations on 6 games, amounting to 864,000 CPU hours and 8,640 V100 GPU hours. The results on the full suite of 57 Atari games required 4,924,800 CPU hours and 49,248 V100 GPU hours. Combining these, we get a total compute budget of 5,788,800 CPU hours and 57,888 V100 GPU hours.

It is worth remembering that the above is likely quite a loose upper-bound, as this estimate assumes 100 percent up time, which is far from the truth given the host-sharing and load-balancing involved in our setup. Additionally, V100 GPUs were chosen based on what was on hand; our models are small enough to fit on much cheaper cards without much slowdown.

## B  COVERAGE METRICS

We calculate two related notions of coverage: lifetime and episodic. Lifetime coverage corresponds to the number of unique states encountered during an agent's lifetime, whereas episodic coverage corresponds to the number of unique states encountered during each episode. Both rely on the notion of a unique state. The subset of games chosen for these metrics were those where a good notion of a unique state is simple: they all involve a controllable avatar that moves in a coordinate system, so unique avatar coordinates are used. This information is exposed in the RAM state of the Atari emulator, as shown in Anand et al. (2019).

## C  ADDITIONAL IMPLEMENTATION DETAILS

For our skill learning reward, one generally helpful change that we had not previously encountered was to clip negative skill rewards to 0 ($r_{skill} = \max(r_{skill}, 0)$). A previous version of this manuscript stated that: "This yields a strictly tighter lower bound on the mutual information (equation 2), since negative rewards imply the discriminator's performance is worse than chance." However, that argument isn't quite correct. Clipping *the expected reward* (i.e. clipping outside the expectation) would produce a tighter bound, but this argument does not hold on a *sample-by-sample* basis (i.e. clipping inside the expectation, as we did). Thus, this should only be viewed as a heuristic.

### REFERENCES

Diederik Kingma and Jimmy Ba. Adam: A method for stochastic optimization. In *International Conference on Learning Representations (ICLR)*, 2015.

| Hyperparameter | Atari | Four Rooms |
|---|:---:|:---:|
| Torso | IMPALA Torso (Espeholt et al., 2018) | tabular |
| Head hidden size | 256 | - |
| Number of actors | 100 | 64 |
| Batch size | 128 | 16 |
| Skill trajectory length ($T$) | 20 | same |
| Unroll length | 20 | same |
| Actor update period | 100 | same |
| Number of skill latents ($N_Z$) | 64 | 128 |
| Replay buffer size | $10^6$ unrolls | same |
| Optimizer | Adam (Kingma and Ba, 2015) | SGD |
| learning rate | $2 * 10^{-4}$ | $2 * 10^{-3}$ |
| Adam $\epsilon$ | $10^{-3}$ | - |
| Adam $\beta_1$ | 0.0 | - |
| Adam $\beta_2$ | 0.95 | - |
| RL algorithm | $Q(\lambda)$ (Peng and Williams, 1994) | same |
| $\lambda$ | 0.7 | same |
| discount $\gamma$ | 0.99 | same |
| Target update period | 100 | - |
| DISDAIN ensemble size ($N$) | 40 | 2 |
| DISDAIN reward weight ($\lambda$) | 180.0 | 10.0 |
| RND reward weight | 0.3 | - |
| Count bonus weight | - | 10.0 |

Table 1: **Hyperparameters.** Atari hyperparameters were tuned on a subset of 6 games (`beam_rider`, `breakout`, `pong`, `qbert`, `seaquest`, and `space_invaders`). In both environments, all RL and skill discovery hyperparameters were tuned for unbonused skill learning and then held fixed when adding exploration bonuses.

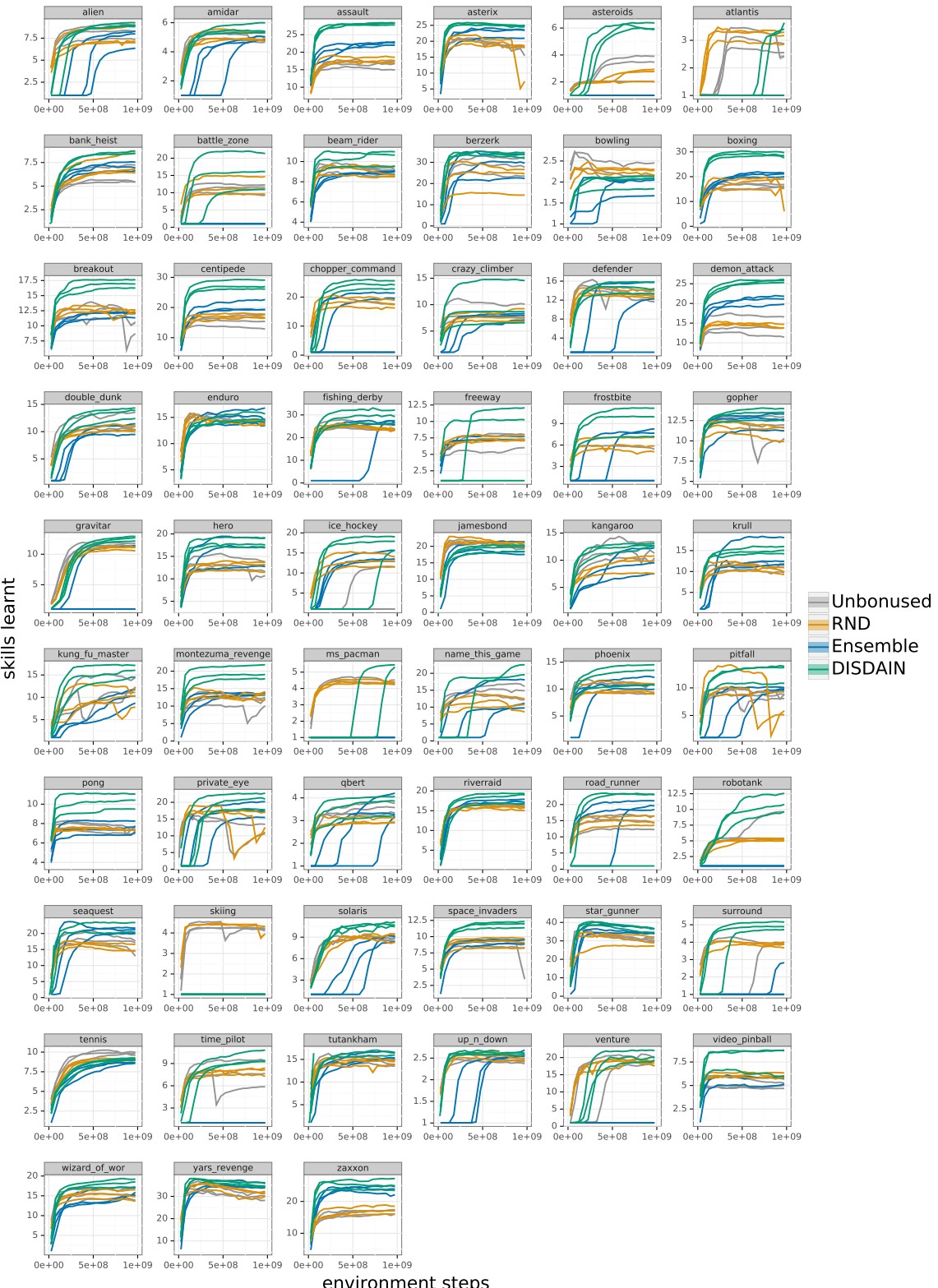

Figure 6: **Skills learnt per-seed and per-game on all 57 Atari games.**

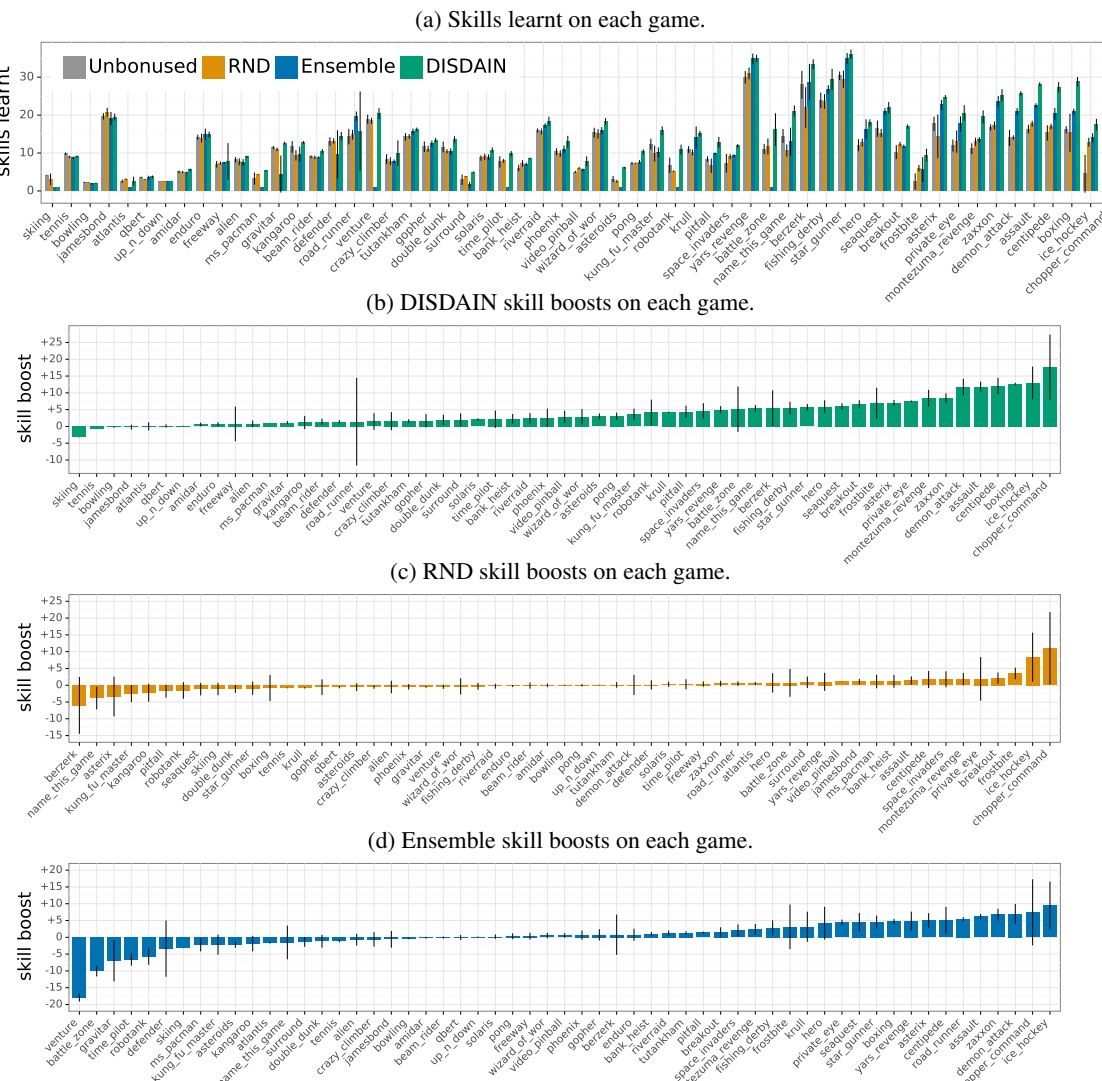

Figure 7: **Per-game boosts for each method.** (b-d) Boosts in skills learnt on each game over Unbonused skill learning. Mean ± standard deviation over 3 seeds. All 3 plots use the same y-axis range magnitude so that bar heights are comparable. DISDAIN improves skill learning on 52/57 (91%) of games, with boosts of >5 skills on 18/57 (32%) and >10 on 6/57 (11%) of games. RND and the Ensemble-only ablation perform closer to chance with improvements on 28/57 (49%) and 35/57 (61%), with boosts of >5 skills on 2/57 (4%) and 9/57 (16%) and >10 skills on 1/57 (2%) and 0/57 (0%) of games, respectively. a) Skills learnt on each game for each method for reference. Sorted by magnitude of DISDAIN boost, as in (b).

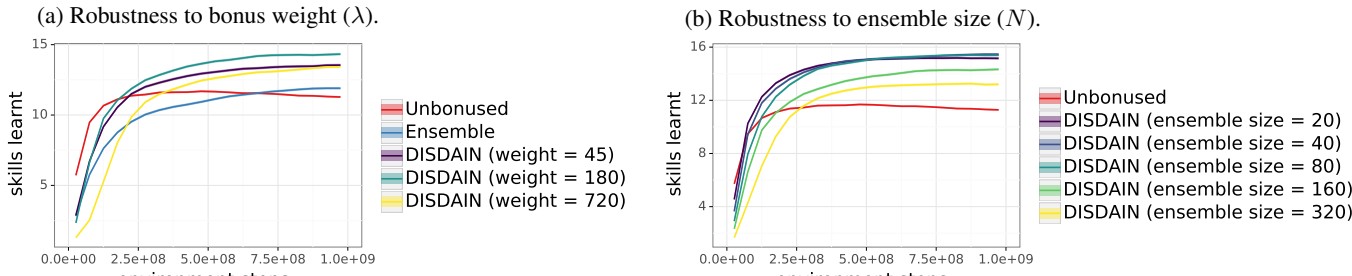

Figure 8: **DISDAIN robustness to key hyperparameters.** (a) Sweeping bonus weight ($\lambda$) with fixed ensemble size $N = 160$ (see Algorithm 1). (b) Sweeping ensemble size ($N$) with fixed bonus weight $\lambda = 180$. Curves averaged over 57 games and 3 seeds (for results broken out by game and seed, see figures 9 and 10). For both hyperparameters, DISDAIN's improvements over baselines are robust over more than an order of magnitude of variation. All other experiments use $\lambda = 180$ and $N = 40$ unless otherwise stated.

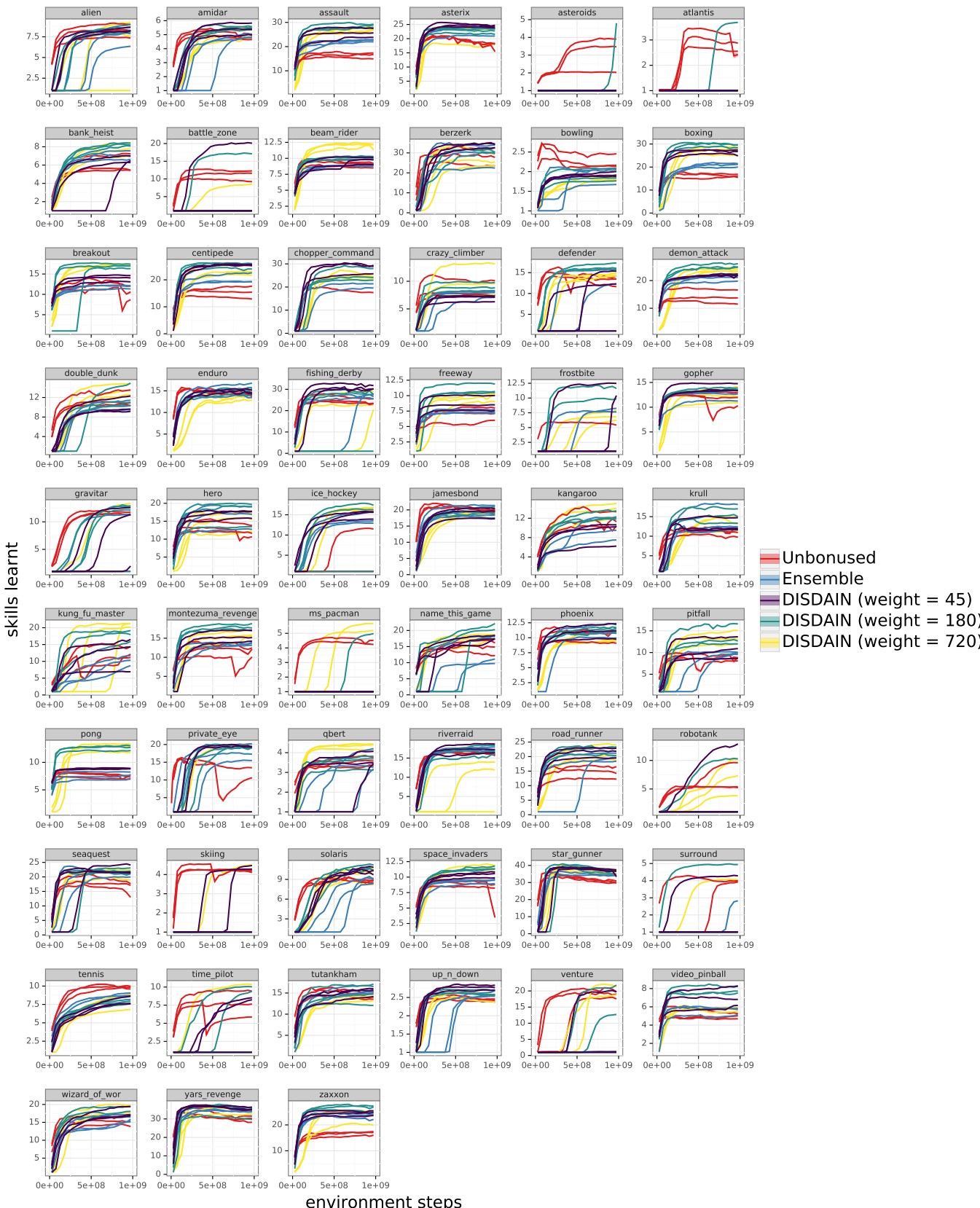

Figure 9: **Per-game bonus weight sweep for DISDAIN across all 57 Atari games.**

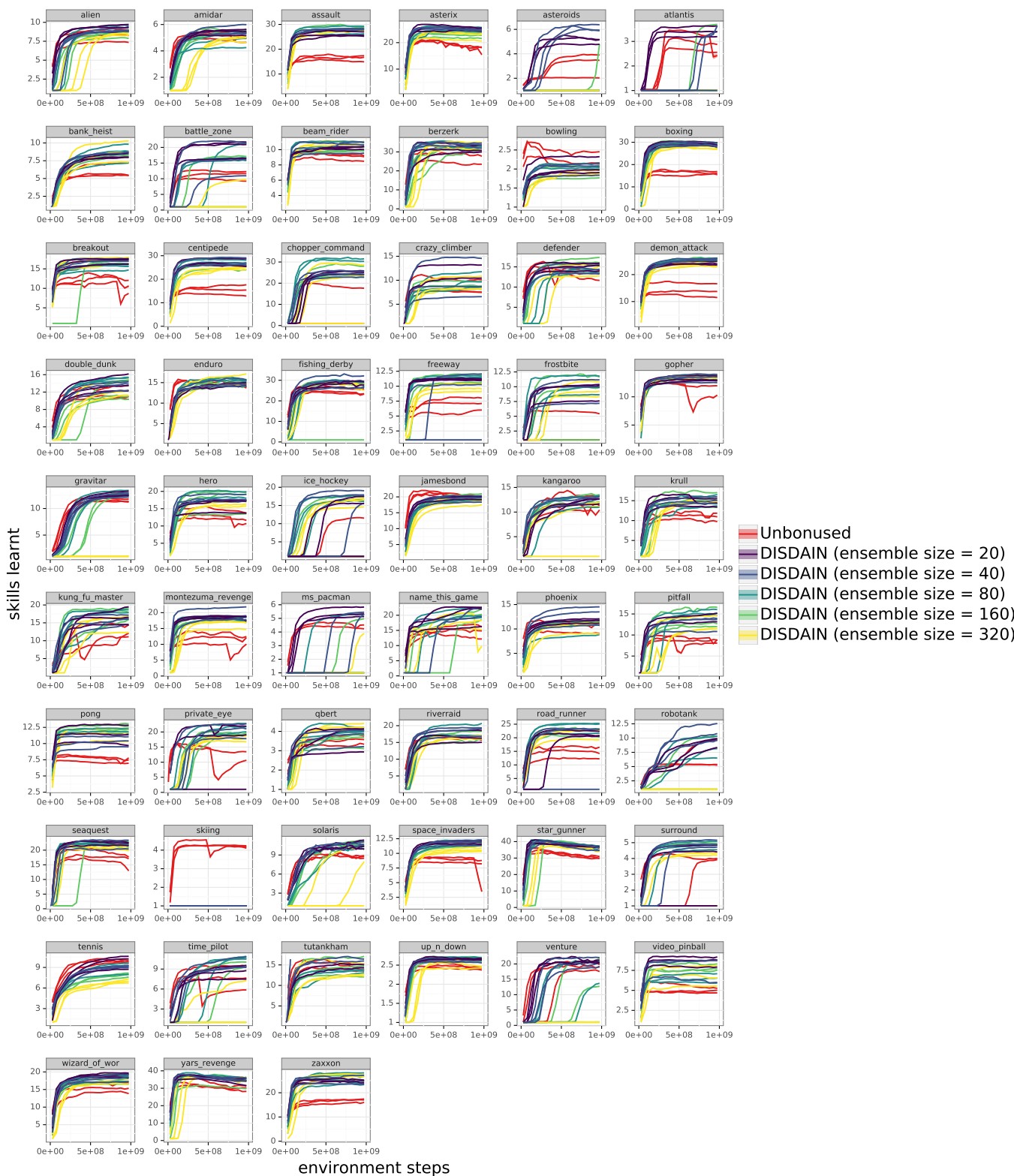

Figure 10: **Per-game ensemble size sweep for DISDAIN across all 57 Atari games.**

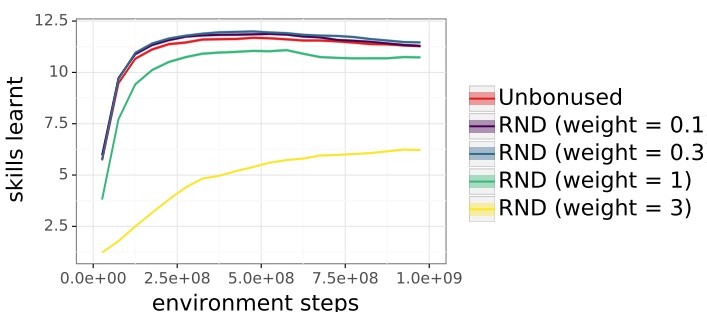

Figure 11: **RND bonus weight sweep.** RND fails to significantly improve skill learning as the weighting on its contribution to the reward is increased. As soon as the RND bonus becomes of a similar order of magnitude as the skill learning reward (bonus weight $\lambda \approx 1$), skill learning performance begins to decay, suggesting that the kind of exploration encouraged by RND is not conducive to skill learning. Results averaged over 57 games and 3 seeds (see figure 12 for results broken out by game and seed).

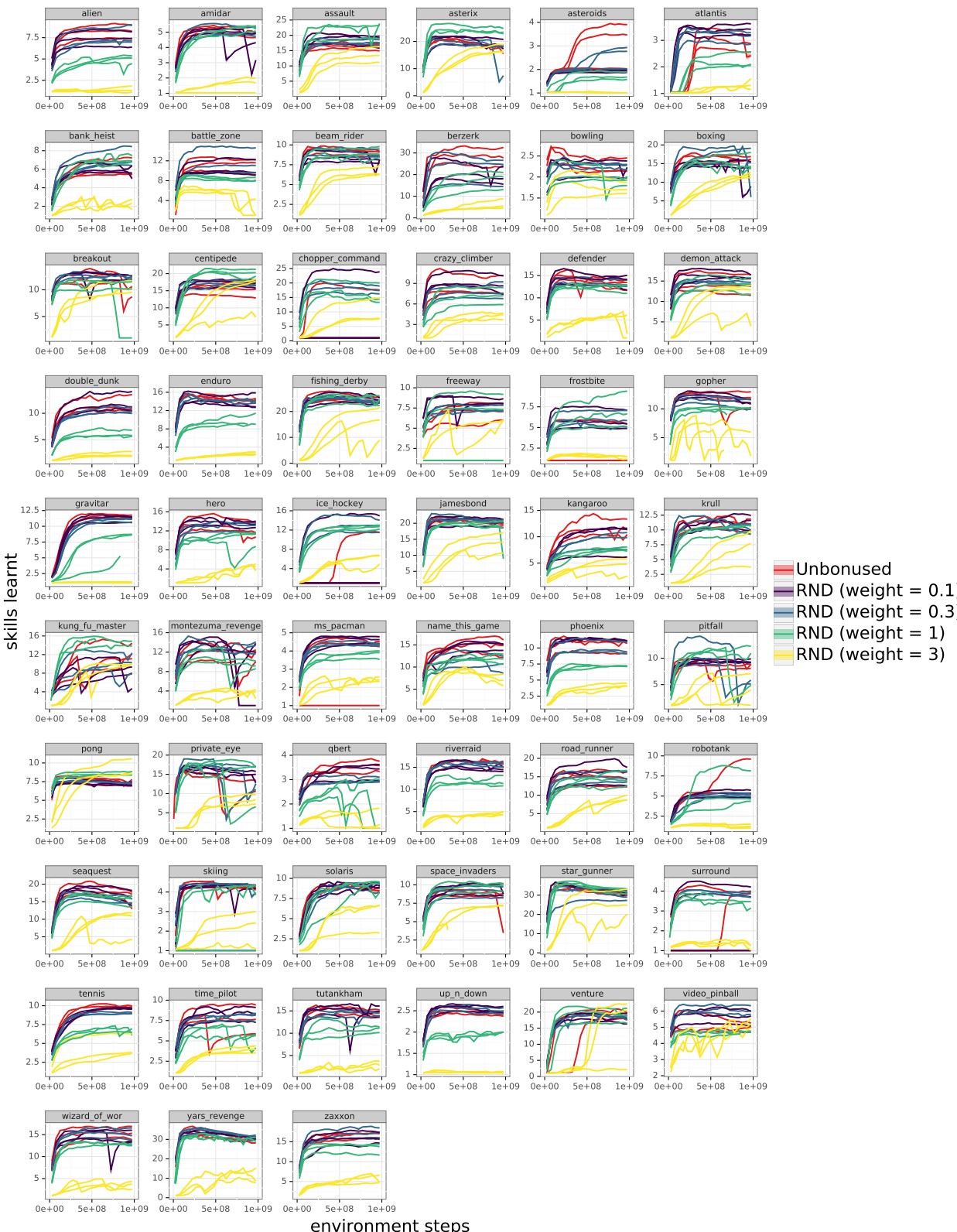

Figure 12: **Per-game bonus weight sweep for RND across all 57 Atari games.**

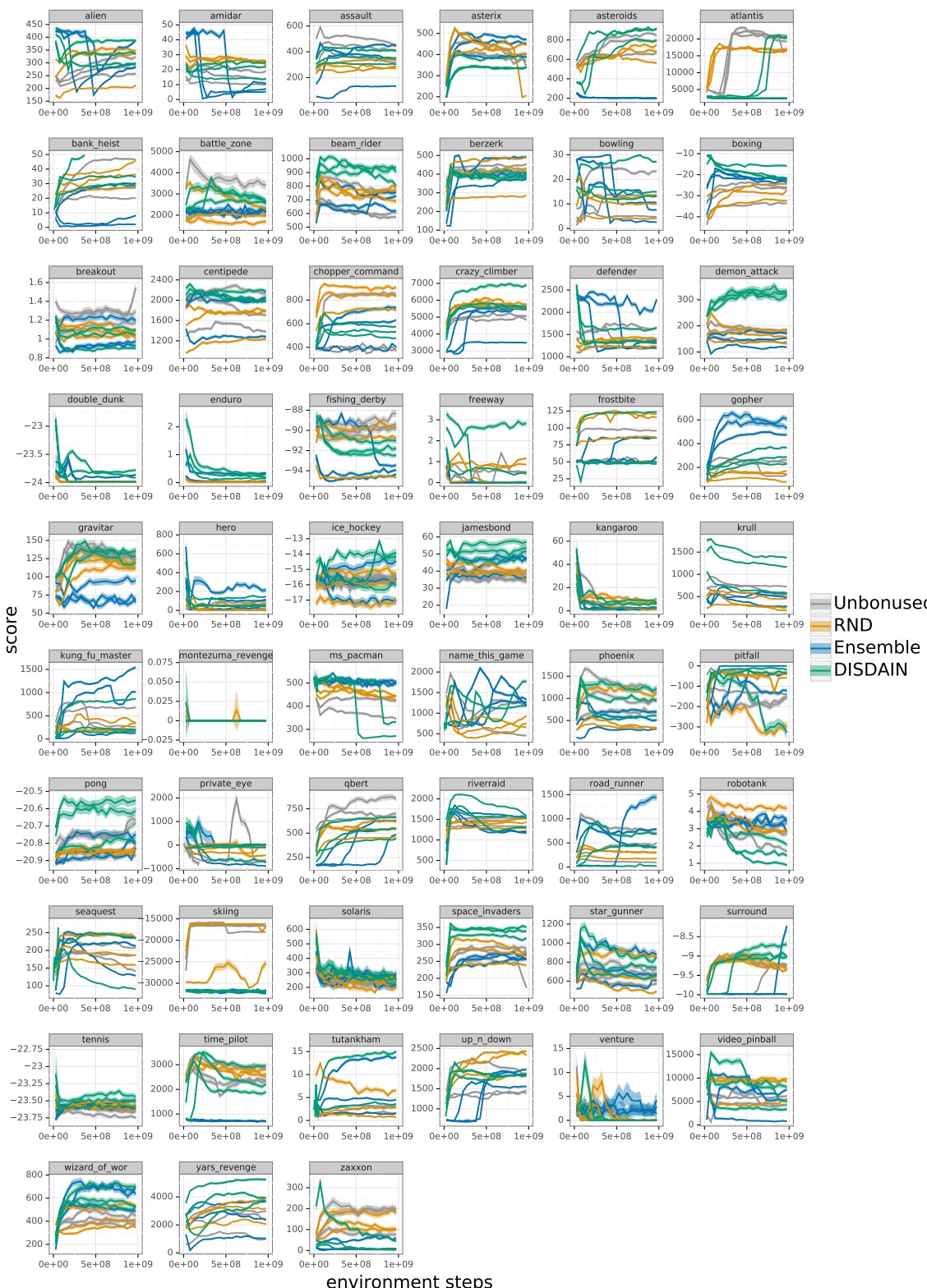

Figure 13: **Per-game task reward attainment curves throughout training.** We emphasize that agents are trained only to maximize the skill learning objective and any associated exploration bonus (i.e. DISDAIN or RND), and not the task reward. Thus, these plots depict zero-shot reward attainment while uniformly randomly switching between skills.

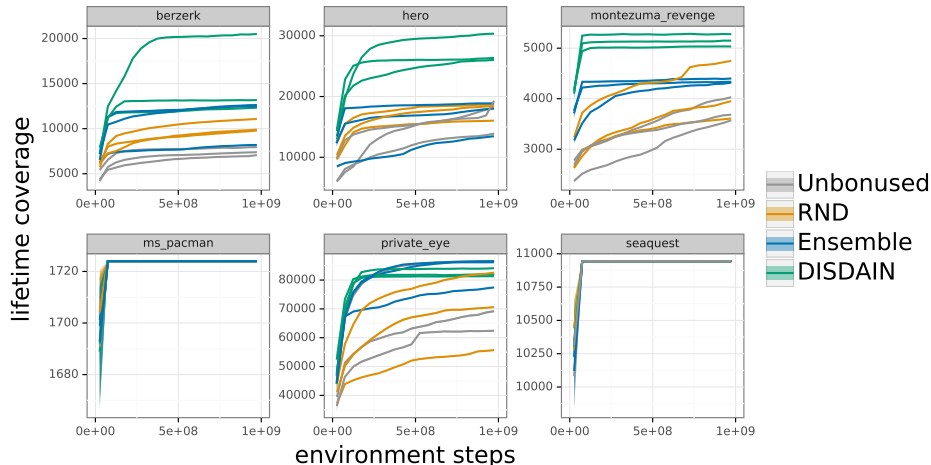

Figure 14: **Lifetime coverage for all games, methods, and seeds.** All policies quickly achieve the same score on ms_pacman and seaquest, so those levels are removed from the analysis in the main text.

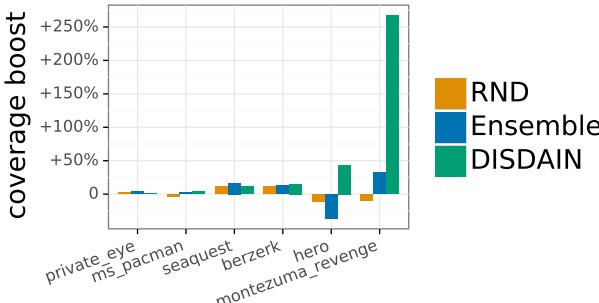

Figure 15: **Episodic coverage boosts over unbonused skill learning.** DISDAIN provides significant boosts over unbonused skill learning on hero and montezuma_revenge, while all methods perform similarly on the other four levels analyzed. Since state coverage metrics are particularly well suited to montezuma_revenge, the boost there is especially interesting. Results averaged over 3 seeds. For results over training for each seed, see figure 16.

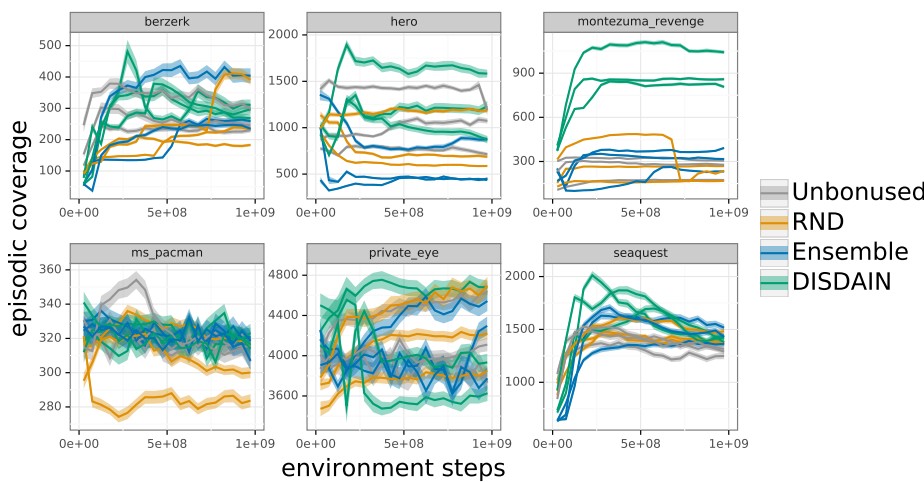

Figure 16: **Episodic coverage for all games, methods, and seeds.**

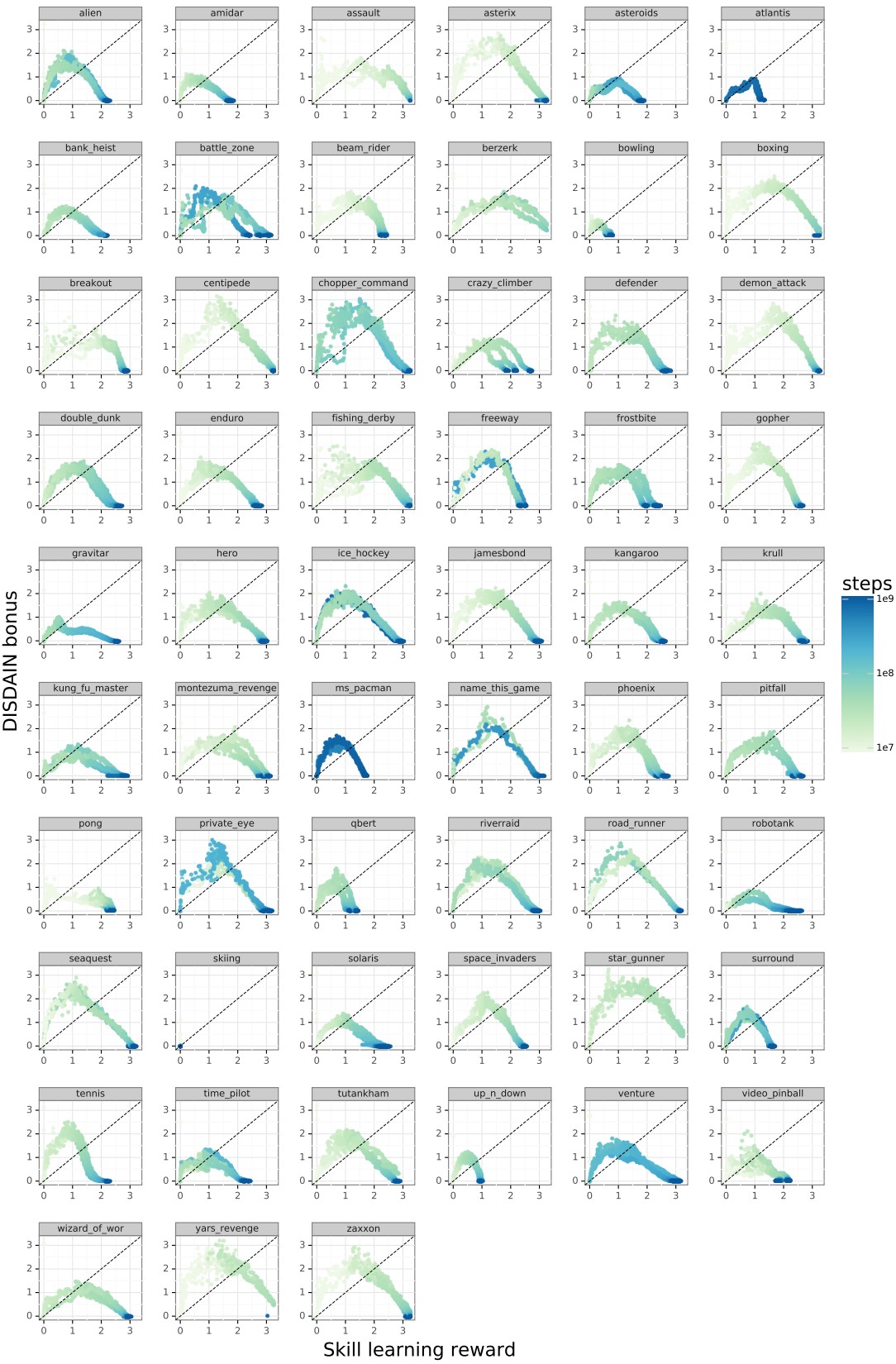

Figure 17: $r_{\text{skill}}$ **vs** $r_{\text{DISDAIN}}$ **during learning for all seeds per-game on all 57 Atari games.** Each panel includes data from 3 seeds. DISDAIN reward tends to dominate early but fades away as skill learning converges.

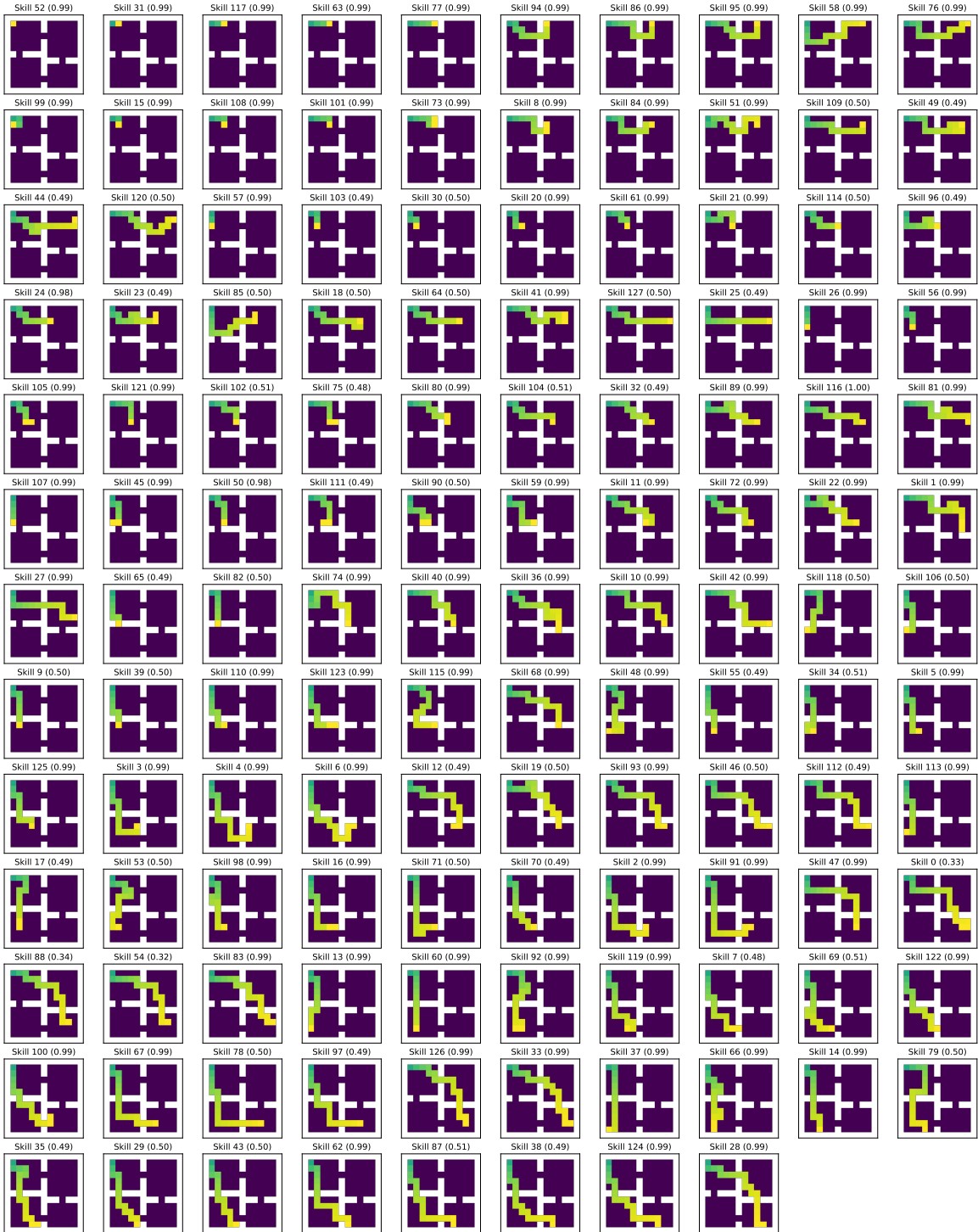

Figure 18: **Example DISDAIN skill rollouts on Four Rooms.** Each panel shows rollout for one skill. Panels are labeled with skill index and the probability the discriminator assigns to the correct skill label. Color indicates time within episode, moving from green (beginning) to yellow (end). Skills are sorted by final state. DISDAIN learns to visit nearly every accessible state.

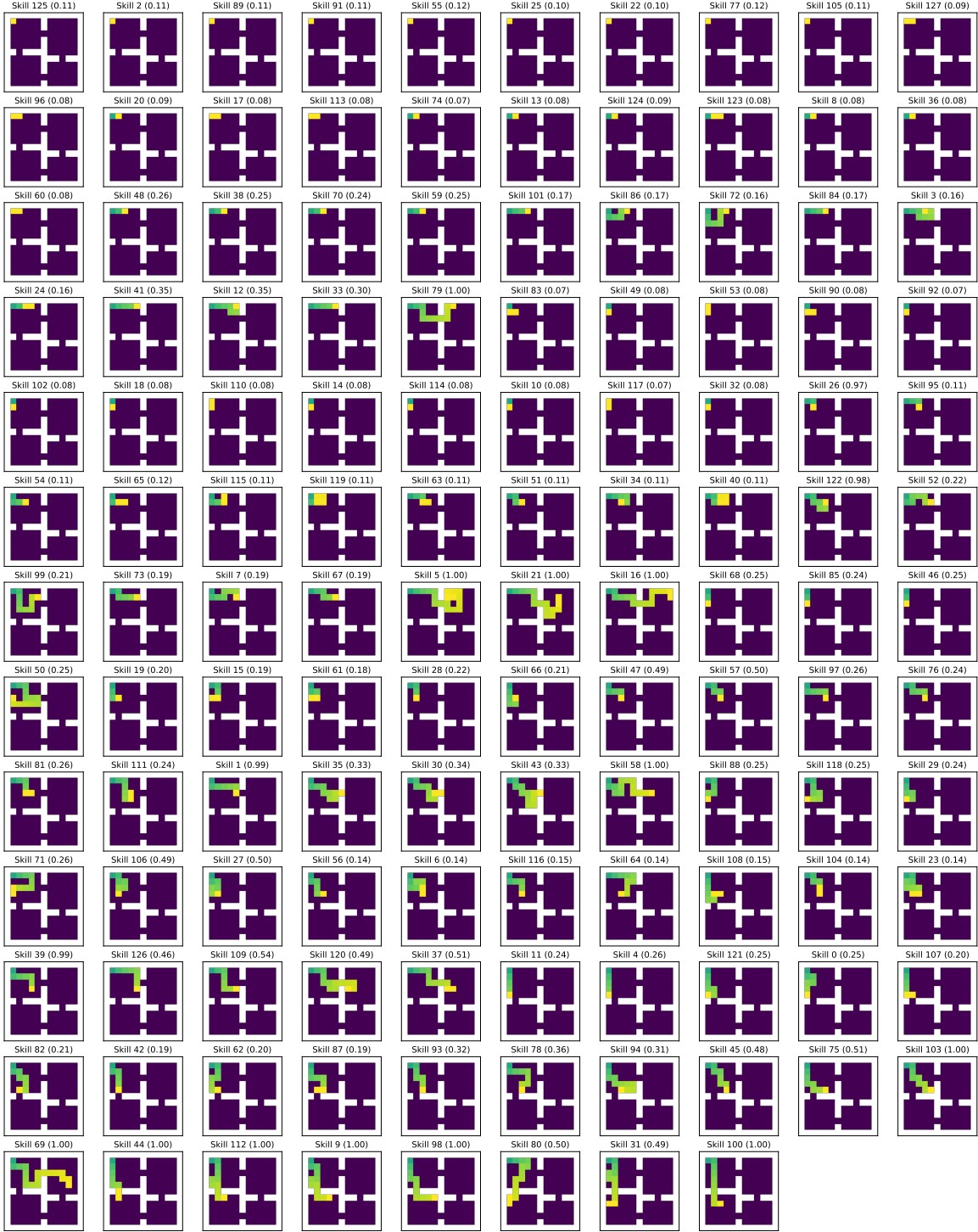

Figure 19: **Example Unbonused skill rollouts on Four Rooms.** Each panel shows rollout for one skill. Panels are labeled with skill index and the probability the discriminator assigns to the correct skill label. Color indicates time within episode, moving from green (beginning) to yellow (end). Skills are sorted by final state. Unbonused skill learning fails to learn to reach many states beyond the first (upper left) room.

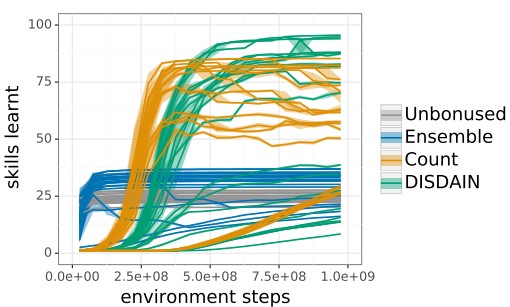

Figure 20: **Four Rooms training curves broken out by seed.** The exploration bonuses dramatically improve skill learning in most cases, though they also slow learning and add variance due to the training of an additional separate value function (and for DISDAIN, an ensemble of discriminators).

