# OpenReview forum: "Learning more skills through optimistic exploration"
_ICLR.cc/2022/Conference — ICLR 2022 Spotlight_

### Official Review · Reviewer_Upj9 · 2021-10-31

**Correctness:** 3
**Technical Novelty And Significance:** 2
**Empirical Novelty And Significance:** 2
**Recommendation:** 6
**Confidence:** 4

**Main Review:**

Strength:
The strengths are summarized below:

1. The writing flow of the paper is very nice, including comprehensive background, easy-to-follow methodology. It would be better to pair with some nice figures.
3. The experiments are pretty solid including various settings environments including tabular form and image-based tasks (Atari). Experiments in Atari are very solid.

Weakness:
I would like to raise several questions to the paper:

1. Can the author provide some experimental supports for the argument that the reward provided by the discriminator is pessimistic in face of uncertain states? I am curious how the reward looks like in Eq. 3 when using functional approximation.

2. As also noted in the paper, the idea of using the ensemble method to encourage exploration has been widely employed in the general RL setting. The proposed intrinsic reward is very similar to those adopted previously, despite the setting being unsupervised skill learning. Thus, I think the algorithm itself is not very novel.

3. In terms of experiments, I think the author should add more comparison between the proposed method and more baselines: for example, a direct comparison would be RND + ensemble.

4. I didn't see the learning curves for downstream tasks (maximize reward) of Atari in the appendix. The author can provide the learning curve for (d) in figure 4.

Clarity: The paper is well-written and clear in the flow.

Feedbacks & Questions: Please see details in the weakness.

**Summary Of The Paper:**

This paper adopts the framework of unsupervised skill learning. To solve the problem that the discriminator will have low confidence in the unseen data thus providing a low intrinsic reward, the paper derives an information gain auxiliary objective that involves training an ensemble of discriminators and rewarding the policy for their disagreement. The paper conducts extensive experiments on tabular grid world and 57 games of the Atari Suite.

**Summary Of The Review:**

Overall, I think the paper is well written and has extensive experiments. My main concern is about the novelty of the method. These concerns are pretty important and thus I encourage the author to engage in the discussion period and clarify these if there is any misunderstanding. I am happy to re-evaluate if the author convinces me.

---

> ### Author Response · Authors · 2021-11-19
> **Review Response (1/2)**
>
> Thank you for your time and feedback.
>
> We will address your concerns on a point-by-point basis, before concluding with a summary of relevant changes to the latest draft of the paper.
>
> > Q1: “Can the author provide some experimental supports for the argument that the reward provided by the discriminator is pessimistic in face of uncertain states? I am curious how the reward looks like in Eq. 3 when using functional approximation.”
>
> A1: Since the reward function is derived from an objective which iteratively tightens a lower-bound, it follows that it will be inherently pessimistic unless the bound is tight. Specifically, the first term in Equation 3 (the second term is a constant) is, on average across trajectories, less than or equal to the true conditional probability: E_\tau[q_\phi(z|o)] <= E_\tau[p(z|o)]. Thank you for raising this concern, as we agree that this was not sufficiently clear in the initial draft of the paper. We hope that the latest draft makes this much more transparent and addresses this concern fully.
>
> > Q2: “As also noted in the paper, the idea of using the ensemble method to encourage exploration has been widely employed in the general RL setting. The proposed intrinsic reward is very similar to those adopted previously, despite the setting being unsupervised skill learning. Thus, I think the algorithm itself is not very novel.”
>
> A2: We set out to solve a problem, not invent a novel method. In our research, we noticed a general trend of unsupervised skill learning algorithms learning fewer skills than we expected. Thus, we set out to better understand the exploration problems present in this setting. As other reviewers attested to, our identification and characterization of these problems alone is a useful contribution. Armed with this understanding, we identified a direct solution that leverages existing off-the-shelf methods. Rather than diminish the significance of our work, we believe this actually enhances it by providing 1) increased confidence in the theoretical soundness and empirical effectiveness of the approach and 2) a literature to potentially source further tools and findings for improving the approach.
>
> That said, we believe that the unsupervised skill learning setting we focus on also has a unique feature that makes uncertainty-based exploration bonuses especially appropriate, namely a learnt reward function. In the other RL settings in which ensemble-based uncertainty bonuses have been used (i.e. value function and forward model uncertainty), the reward function (i.e. task reward) is fixed and independent of the policy. Here, exploration only needs to sufficiently cover the state space, and extra exploration only hurts to the extent that it affects data efficiency. Hence why much subsequent work has been able to replace ensemble-based uncertainty bonuses with other state exploration methods (e.g. RND). In contrast, in the unsupervised skill learning setting that we treat, the reward function (i.e. skill learning reward) is non-stationary and policy-dependent (that, the reward function is learnt). Here, exploration must be conducted carefully and in a targeted fashion, since it in turn alters the learnt reward function (via the discriminator). Thus an ensembled-based uncertainty bonus is especially well suited to this setting.
>
> > Q3: “In terms of experiments, I think the author should add more comparison between the proposed method and more baselines: for example, a direct comparison would be RND + ensemble.”
>
> A3: We agree that more baselines are always better, but note that these are expensive in terms of time and cost. The ensemble baseline does seem to yield some benefits, but as the ensemble is just a change in the predictor’s architecture (i.e. it is not being used for information gain calculations in this condition), these benefits are limited to things like a better weight initialization or gradient propagation. On the other hand, the RND baseline has failed to provide any significant benefit. And this is true despite significant hyper-parameter tuning (see Appendix Figure 10). Indeed, any significant RND weight actually hurts performance, which provides evidence for our hypothesis that this objective is not well aligned with that of skill learning. If there is reason to believe this might be remedied by changing the architecture of the predictor to be an ensemble, then we will be happy to try to run this experiment. However at present we fail to see the intuition behind this request.
>
> (Continued below)

---

> > ### Author Response · Authors · 2021-11-19
> > **Review Response (2/2)**
> >
> > > Q4: “I didn't see the learning curves for downstream tasks (maximize reward) of Atari in the appendix. The author can provide the learning curve for (d) in figure 4.”
> >
> > A4: We would like to clarify that the referenced figure (Figure 4d in the original submission; Figure 5a in the updated draft) is the task reward obtained while training to maximize the unsupervised objective, not while directly maximizing task reward. Thus, it is a “zero-shot” measure of reward attainment meant to assess the utility of our learnt skills “out of the box” without further optimization. That said, we have added curves showing the task reward attainment throughout learning to the appendix (Figure 13).
> >
> > ## Summary of changes
> > - As noted in A1, we have amended the text to make the pessimism of the objective more transparent.
> > - As noted in A4, we have added “task reward attainment throughout learning” curves to the appendix (Figure 14).
> > - We have increased the rigor of our Atari evaluation by switching our analysis plots to a more statistically robust format, including appropriately bootstrapped error bars (per [1] and the OpenReview public comment from the first author). The new evaluations reaffirm the statistically significant performance increases of DISDAIN over baselines. (Please note that we will update the color and style of our remaining plots to be consistent with these before the camera-ready deadline, as well as add the requisite details of the statistical analysis to a new section in the appendix.)
> > [1] Agarwal et al, ​​Deep reinforcement learning at the edge of the statistical precipice, NeurIPS 2021
> >
> > In light of our responses to all of your concerns, and the substantial improvements listed above, we hope you’ll consider raising your score to acceptance.

---

> ### Comment · Reviewer_Upj9 · 2021-11-23
> **Response**
>
> I would like to thank the authors for their detailed response and revision of the paper. I feel at this point, my questions are all been addressed. I am raising my score accordingly.

---

### Official Review · Reviewer_7dHZ · 2021-11-01

**Correctness:** 4
**Technical Novelty And Significance:** 4
**Empirical Novelty And Significance:** 4
**Recommendation:** 8
**Confidence:** 5

**Main Review:**

## Strengths:
- An important problem in unsupervised skill discovery through info-max methods (such as VIC and DIAYN) is identified.
- A simple and intuitive approach is proposed to fix the issue.
- The problem and the solution method are both well-motivated.
- Experiments (featuring a standard toy problem --- Four Rooms domain --- as well as pixel-based Atari domains) show strong support for the effectiveness of the solution method.
- Justifiable sets of baselines are used in each environment.
- Presentation is clear and provides a good summary of prior work.

## Weaknesses:
- Discussion of limitations is only included in the supplementary material and should be brought to the main text and discussed earlier on.
- Given the higher variability per seed, why do none of the plots shows variance across seeds?
- Also, not sure why only 3 seeds are run in Atari? Considering this higher variability, shouldn't a larger number of seeds be used? For DQN, the standard is 5 seeds and DQN already shows very low variability across seeds.
- It is unclear how the authors have realized the sufficiency of not using bootstrapping for training the ensemble (i.e. they use the same mini-batches across the ensemble) --- see my Question 1. Also, not clear how much this choice is responsible for higher variability across seeds?
- It would have been nice to see a qualitative analysis of learned skills in Atari. Is that not something that can be done quite easily?


## Additional questions:
1. On page 4, it is stated that the same mini-batches are used to train the discriminators within the ensemble and that this was found to be "*both sufficient and simpler*". Regarding sufficiency: (i) how did you experiment to find out bootstrapping vs. same mini-batches yield similar performances? (ii) wouldn't this, in the long run, be problematic/limiting (in a general sense)?

2. In Sec. 5 it is mentioned that this work is the first to apply ensemble learning and epistemic uncertainty capturing to unsupervised skill discovery. However, curiosity has been combined with ensemble models before (Pathak et al., 2019). Also, curiosity has been used to acquire skills by snapshotting (Ref. [1]). While I agree that in some senses the statement in the paper is reasonable, I think this general statement confuses more than it informs. Wouldn't it be more clear if this was stated in the specific context of *diversity-seeking* unsupervised skill discovery?

## Minor:
- Fig. 4: Legend seems out of place (it's included in part (b), while it's least useful in the context of part (b)).
- Colors in Fig. 4 are too close (RND and Ensemble are difficult to distinguish in part (a)).

### References:
[1] Groth, O. et al. (2021). Is Curiosity All You Need? On the Utility of Emergent Behaviours from Curious Exploration. *arXiv 2109.08603*.

**Summary Of The Paper:**

This paper identifies a source of pessimism in DIAYN-style methods for exploring new parts of the state space. They argue that this issue is due to using a single point estimator as a discriminator and that capturing the epistemic uncertainty of the discriminator could serve as an additional signal to guide exploration. They achieve this by using an ensemble of discriminators and incorporating the epistemic uncertainty across the ensemble into an additional intrinsic reward to the diversity of skills reward (through a mixing parameter $\lambda$). They examine this method against DIAYN-style methods and count-based methods and show that this new approach broadly outperforms both these classes of methods.

**Summary Of The Review:**

The paper introduces and solves an important problem, proposes a technically sound method, has a very clear presentation of related works, ideas, and results, and offers good experimental evidence for the claims.

Nevertheless, I believe certain clarifications/additions could improve the paper.

---

> ### Author Response · Authors · 2021-11-19
> **Review Response (1/2)**
>
> Thank you for your time and consideration. We are particularly pleased by your assessment that our paper’s “problem and the solution method are both well-motivated”. But we also want to address your more constructive feedback, and we have done this on a point-by-point basis, with a summary of relevant textual changes at the end.
>
> > Q1: “Discussion of limitations is only included in the supplementary material and should be brought to the main text and discussed earlier on.”
>
> A1: We agree! Done.
>
> > Q2: “Given the higher variability per seed, why do none of the plots shows variance across seeds? Also, not sure why only 3 seeds are run in Atari? Considering this higher variability, shouldn't a larger number of seeds be used? For DQN, the standard is 5 seeds and DQN already shows very low variability across seeds.”
>
> A2: We’re not certain that the methods considered here do have higher variability per seed (compared to DQN?), and perhaps the closest point of comparison would be the VISR paper which ran similar skill discovery methods on Atari with 3 seeds. That said, we have added bootstrapped confidence intervals to all of the results. Additionally, we now provide ‘performance profiles’ and ‘chance of improvement’ plots that aggregate over all games (171 seeds). Both of these measures (suggested by Rishabh Agarwal via public OpenReview comment and in his paper [1]) show highly significant (i.e. outside 95% bootstrapped confidence interval) benefits of DISDAIN over all baselines. We hope these changes and additions address your underlying concerns around the statistical significance of our results.
>
> > Q3: “It is unclear how the authors have realized the sufficiency of not using bootstrapping for training the ensemble (i.e. they use the same mini-batches across the ensemble) --- see my Question 1. Also, not clear how much this choice is responsible for higher variability across seeds?” and “On page 4, it is stated that the same mini-batches are used to train the discriminators within the ensemble and that this was found to be "both sufficient and simpler". Regarding sufficiency: (i) how did you experiment to find out bootstrapping vs. same mini-batches yield similar performances? (ii) wouldn't this, in the long run, be problematic/limiting (in a general sense)?”
>
> A3: We did some initial experimentation with this in the early stages of the project, but found no significant advantage so we simplified the method accordingly. Indeed, the Bootstrapped DQN paper [2] came to the same conclusion (they use ensembles for value function uncertainty estimation): “Figure 5b sweeps over data sharing between ensemble members and finds it has little to no effect.” and “Bootstrapped DQN relies upon random initialization of the network weights as a prior to induce diversity. Surprisingly, we found this initial diversity was enough to maintain diverse generalization to new and unseen states for large and deep neural networks.” That said, we note that DISDAIN should be compatible with most enhancements to the basic “ensembles for uncertain estimation” formula, so it might be worth revisiting such techniques when applying DISDAIN to even more complex environments.
>
> > Q4: “It would have been nice to see a qualitative analysis of learned skills in Atari. Is that not something that can be done quite easily?”
>
> A4: We agree in principle, but we have not yet found a scalable way to provide such analysis across the full task suite and we believe that showing individual seeds/games is unlikely to add significant value (e.g. it is prone to cherry-picking). We believe developing such analyses would represent a novel contribution in its own right, and is a promising direction for future work. We hope that the coverage and task reward statistics help address this issue by providing more information about skill behavior (e.g. random skill execution tends to explore the state space).

---

> > ### Author Response · Authors · 2021-11-19
> > **Review Response (2/2)**
> >
> > > Q5: “In Sec. 5 it is mentioned that this work is the first to apply ensemble learning and epistemic uncertainty capturing to unsupervised skill discovery. However, curiosity has been combined with ensemble models before (Pathak et al., 2019). Also, curiosity has been used to acquire skills by snapshotting (Ref. [1]). While I agree that in some senses the statement in the paper is reasonable, I think this general statement confuses more than it informs. Wouldn't it be more clear if this was stated in the specific context of diversity-seeking unsupervised skill discovery?”
> >
> > A5: We acknowledge that both of these works are related to our own and that descriptive terminology is important. That said, (Pathak et al 2019) doesn’t use ensemble models to learn skills (i.e. conditional policies), and while Ref. [1] does learn skills, it doesn’t use ensembles and it first appeared online after our initial submission. As such, we believe that our statement is correct, but will happily modify it if we missed something in our read through of these papers.
> >
> > Summary of changes:
> > - We have moved the limitations section into the main paper (from the appendix).
> > - We have increased the rigor of our Atari evaluation by switching our analysis plots to a more statistically robust format, including appropriately bootstrapped error bars (per [1] and the OpenReview public comment from the first author). The new evaluations reaffirm the statistically significant performance increases of DISDAIN over baselines. (Please note that we will update the color and style of our remaining plots to be consistent with these before the camera-ready deadline, as well as add the requisite details of the statistical analysis to a new section in the appendix.)
> >
> > [1] Agarwal et al, ​​Deep reinforcement learning at the edge of the statistical precipice, NeurIPS 2021
> > [2] Osband et al, Deep exploration via bootstrapped DQN, NeurIPS 2016

---

> > > ### Comment · Reviewer_7dHZ · 2021-11-20
> > > **Thanks for your response!**
> > >
> > > Thanks for addressing my concerns, especially the one about statistical significance which was my most major concern. I am now happy to keep my score of 8 (which I had originally given assuming that the concerns would be addressed) and advocate for accepting the paper. Additionally, I have now increased my scores for Correctness and Confidence.

---

### Official Review · Reviewer_rscN · 2021-11-02

**Correctness:** 3
**Technical Novelty And Significance:** 3
**Empirical Novelty And Significance:** 3
**Recommendation:** 8
**Confidence:** 3

**Main Review:**

**Strengths**

The paper is well-written and I enjoyed reading it. The authors clearly explain the discriminator-based skill discovery via intrinsic rewards given in (3) and why it results in a pessimistic agent. In skill discovery, a low reward is given when the discriminator is poor which happens in the face of unseen states. This is contradictory to the UFO principle for exploration and thus motivates the authors to provide an auxiliary bonus. The discussion on page 4 regarding maximizing the lower bound and ensuring its tightness is interesting. The new exploration method is simple and practical and is shown empirically to improve skill discovery.

**Limitations/Comments**

The problem considered in this paper is closely related to the reward-free pure exploration setting. For that setting, approaches based on maximum entropy exploration have been proposed, such as

Zhang, Chuheng, Yuanying Cai, and Longbo Huang Jian Li. "Exploration by Maximizing R\'enyi Entropy for Reward-Free RL Framework." Proceedings of the AAAI Conference on Artificial Intelligence. Vol. 35. No. 12. 2021.

Comparison with these methods is not provided in the paper.

As the authors acknowledge, it is not clear whether the number of learned skills in (4) is a good performance metric. A better metric can be similar to the reward-free paradigm, which is being able to solve any task if the reward function is provided.

In empirical evaluations, the authors compare with RND which is a common pseudocount-based method. However, several other exploration methods are proposed that significantly improve over RND:

Seo, Younggyo, et al. "State entropy maximization with random encoders for efficient exploration." arXiv preprint arXiv:2102.09430 (2021).

Raileanu, Roberta, and Tim Rocktäschel. "RIDE: Rewarding impact-driven exploration for procedurally-generated environments." arXiv preprint arXiv:2002.12292 (2020).

Zhang, Tianjun, et al. "MADE: Exploration via Maximizing Deviation from Explored Regions." arXiv preprint arXiv:2106.10268 (2021).

**Summary Of The Paper:**

This paper is concerned with unsupervised RL where an extrinsic reward signal is not available. The objective is for the agent to master the environment by exploring it while learning a diverse set of skills. This is done by simultaneously training a policy conditioned on a latent variable and a discriminator that tries to infer the latent variable from trajectories. The authors identify that the intrinsic rewards used in skill discovery result in the agent being pessimistic towards exploring novel parts of the environment. To alleviate this, the authors propose a new auxiliary objective, which results in a bonus based on the disagreement of an ensemble of discriminators. Empirical results on the grid world and Atari show improvements in skill discovery and solving downstream tasks compared to baselines.

**Summary Of The Review:**

The motivation behind the method is well-justified and the approach is interesting and practical. The discussion on pessimism in skill discovery is insightful. The paper can be improved by further comparison with methods that are not focused on skill discovery.

---

> ### Author Response · Authors · 2021-11-19
> **Review Response**
>
> Thank you for your time and very considerate review. We are particularly pleased by your description of DISDAIN as “simple and practical”. But we also want to address your more constructive feedback, and we have done this on a point-by-point basis, with a summary of relevant textual changes at the end.
>
> > Q1: “The problem considered in this paper is closely related to the reward-free pure exploration setting. For that setting, approaches based on maximum entropy exploration have been proposed, such as [Exploration by Maximizing R'enyi Entropy for Reward-Free RL Framework]. Comparison with these methods is not provided in the paper.”
>
> A1: We agree that the problem setting is quite related to reward-free pure exploration, and that there are many different techniques to address this challenge. That said, the aim of this paper is to address a limitation in one such line of work, namely the pessimistic rewards inherent to unsupervised skill discovery techniques. The larger question of whether or not skill discovery is better than e.g. state entropy maximization for a partial use-case is an important one, but one that lies outside the scope of this work.
>
> > Q2: “As the authors acknowledge, it is not clear whether the number of learned skills in (4) is a good performance metric. A better metric can be similar to the reward-free paradigm, which is being able to solve any task if the reward function is provided.”
>
> A2: Related to the previous point, this comes down to a conscious choice to limit the scope of this work, in order to increase the robustness of the results (i.e. fewer evaluations allow for more seeds and hyper-parameter sweeps). In particular, the class of methods we aim to improve (unsupervised skill discovery) have not yet converged on a single dominant downstream use-case, e.g. RL vs imitation learning vs goal achievement. Earlier work (e.g. DIAYN) has at least shown that unsupervised skill discovery is capable of all of these use cases, and thus we view improvements to the class of algorithms as worthy in their own right.
>
> > Q3: “In empirical evaluations, the authors compare with RND which is a common pseudocount-based method. However, several other exploration methods are proposed that significantly improve over RND…”
>
> A3: We chose RND largely based on the extent to which it has been repeatedly demonstrated on the full Atari suite. To the best of our understanding, none of the papers referenced ran across all 57 games. While their initial results suggest that this would not be problematic, having published hyper-parameters that work across the suite improves RND’s utility as a baseline (i.e. there is less question as to whether the method was sufficiently adapted to the environment at hand). That said, our latest draft now cites these papers and notes the rationale behind our decision to choose RND.
>
> Summary of changes:
> - Q1 and Q2 mentioned in an expanded limitations section i.e. comparisons between exploration and skill discovery methods are necessary, as are downstream evaluations
> - Justification for choice of RND given in the paper, and the alternatives mentioned in Q3 are noted
> - We have increased the rigor of our Atari evaluation by switching our analysis plots to a more statistically robust format, including appropriately bootstrapped error bars (per [1] and the OpenReview public comment from the first author). The new evaluations reaffirm the statistically significant performance increases of DISDAIN over baselines. (Please note that we will update the color and style of our remaining plots to be consistent with these before the camera-ready deadline, as well as add the requisite details of the statistical analysis to a new section in the appendix.)
>
>
> [1] Agarwal et al, ​​Deep reinforcement learning at the edge of the statistical precipice, NeurIPS 2021

---

> > ### Comment · Reviewer_rscN · 2021-11-26
> > **Response**
> >
> > I thank the authors for providing a detailed response to all the concerns raised by the reviewers. I am happy with their response and keep my original score of 8, voting for acceptance of this paper.

---

### Official Review · Reviewer_vjRd · 2021-11-03

**Correctness:** 4
**Technical Novelty And Significance:** 3
**Empirical Novelty And Significance:** 3
**Recommendation:** 8
**Confidence:** 5

**Main Review:**

## Strengths

1. The paper is well motivated. The shortcomings of existing unsupervised skill discovery algorithms are clearly explained along with how the proposed method overcomes these shortcomings.

1. The derivation of the proposed objective is intuitively appealing and theoretically sound, the final form of the reward bonus is a simple one (a desirable trait) -- entropy of the mean discriminator minus the mean entropy of the ensemble of discriminators.

1. The paper does a very good job in explaining all the hyperparameter and modeling choices made in their experiments and reproduction of baselines.

1. The paper does a good job at thoroughly analysing the pedagogical 4-room setting as well as all 57 Atari game environments.

1. The results in the 4-room and Atari environments are both positive in terms of the three metrics evaluated -- number of skills learned, transfer of skills to downstream tasks and lifetime state space coverage. Appropriate ablations are compared to in all cases, specifically,

## Weaknesses

1. Unlinke prior work (DIAYN), the paper restricts the choice of environments to the 4-room and Atari environments. Given that DIAYN and subsequent works (e.g. DADS [1]) have shown skill learning in continuous control tasks such as in the MuJoCo locomotion environments, it would be good to see results of the presented experiments in such domains.

## Other comments and feedback

1. Figure 2 does not do a good job in conveying meaningful information about the methodology. For 2(a), the notation is dense and hard to parse, the different colored boxes for each component don’t help. I don’t see the need for Figure 2(b) just to explain the disagreement objective from the ensemble, this was better conveyed in text. Dropping this figure may be a good idea.

### References:

[1] Sharma, A., Gu, S., Levine, S., Kumar, V., & Hausman, K. (2019). Dynamics-aware unsupervised discovery of skills. arXiv preprint arXiv:1907.01657.


================
## Post-rebuttal review update

The authors have addressed the concerns raised by all reviewers and I will maintain my score of 8 (Accept).


**Summary Of The Paper:**

The paper proposes a novel unsupervised skill discovery algorithm. Beginning with the family of such methods (DIAYN, VIC, etc) that learn a discriminator to distinguish skills given some observations of the trajectory and a policy that executes a skill conditioned on the (discrete) skill random variable Z, the premise of the paper is that such methods on their own will fail when new states are encountered during the skill learning process as the discriminator would not have had sufficient data to learn to distinguish novel states. The paper proposes a novel reward bonus that works in addition to a base method such as DIAYN (DIversity is All You Need), such that this bonus “reimburses” the policy for visiting states where the discriminator uncertainty is high (measured using a form of disagreement across an ensemble of discriminators). Experiments on the pedagogical 4-room environment and the Atari suite of environments demonstrates the benefit of the proposed reward bonus in not only learning more skills (or “empowerment” in the VIC nomenclature), but the learnt skills are also superior for downstream tasks (external reward) and lifetime state coverage.

**Summary Of The Review:**

The paper is well motivated, provides a simple and theoretically sound novel reward bonus for overcoming the stated “pessimistic exploration” problem and demonstrates positive results across the board for the 4-room and the 57 Atari game environments. No experiments are presented on continuous control tasks such as MuJoCo locomotion environments, which would have strengthened the paper further.

---

> ### Author Response · Authors · 2021-11-19
> **Review Response**
>
> Thank you for your time and consideration. We particularly appreciate your judging the objective to be “intuitively appealing and theoretically sound”. But we also want to address your more constructive feedback.
>
> > “Unlinke prior work (DIAYN), the paper restricts the choice of environments to the 4-room and Atari environments. Given that DIAYN and subsequent works (e.g. DADS [1]) have shown skill learning in continuous control tasks such as in the MuJoCo locomotion environments, it would be good to see results of the presented experiments in such domains.”
>
> We agree that more domains would further increase the impact of our work, and this is an important direction for future research. This also doesn’t necessarily pose a problem for DISDAIN itself, as the method is agnostic to action and observation spaces. That said, these differences require additional hyper-parameter tuning for e.g. the underlying RL algorithms, which becomes time and computation cost prohibitive considering the scale of our existing experiments. Indeed, this is likely why most skill discovery papers tend to run on either continuous control (e.g. DADS, DIAYN) or Atari (VISR). We also note that DADS and DIAYN interface with these continuous control environments via a relatively low dimensional state representation, as opposed to the pixel-based inputs of Atari. This makes adapting our RND baseline non-trivial, as (to the best of our knowledge) it hasn’t been applied in such settings and any poor performance needs to be attributable to its interaction with skill discovery and not a byproduct of untested hyper-parameters.
>
> While unrelated to your feedback, we thought you might be pleased to know that we have increased the rigor of our Atari evaluation by switching our analysis plots to a more statistically robust format, including appropriately bootstrapped error bars (per [1] and the OpenReview public comment from the first author). The new evaluations reaffirm the statistically significant performance increases of DISDAIN over baselines. (Please note that we will update the color and style of our remaining plots to be consistent with these before the camera-ready deadline, as well as add the requisite details of the statistical analysis to a new section in the appendix.)
>
> [1] Agarwal et al, ​​Deep reinforcement learning at the edge of the statistical precipice, NeurIPS 2021

---

### Public Comment · ~Rishabh_Agarwal2 · 2021-11-11
**Statistical uncertainty in Atari results & Suggestions for reliable evaluation**

Hi authors,

The empirical evaluation can be further strengthened. Looking at the individual runs in the appendix, there is a sizeable statistical uncertainty in results (due to the use of only 3 seeds) but this is not reported in the main Atari results in Figure 4. However, reporting only point estimates can have significant influence in performance comparisons and even lead to erroneous conclusions on widely-used benchmarks including Atari 2600 games [1].

Since you have access to individual runs, here are some suggestions to report results reliably using the fact that you have access to scores on 57 tasks x 3 seed/tasks = 171 seeds. All of the above suggestions for Figure 4 can be easily incorporated using the library at https://github.com/google-research/rliable or the [colab](https://bit.ly/statistical_precipice_colab).

 - (a) The bar plot needs to have error bars for each of the games. However, such bar plots are hard to read due to per game differences and error bars would make results.  A better alternative would be **performance profiles** with bootstrap confidence intervals which would show the distribution of skills learnt combined across all games and runs. Such curves allow comparisons at a glance: area under the curve corresponds to mean score, any percentile can be easily read from such plot while one curve above another imply stochastic dominance.

 - *Confidence intervals* need to be reported for the aggregate statistics in (c). Stratified bootstrap CIs make use of the entire 171 seeds and can be used for this purpose.

- Also, note that mean is prone to outliers while median can result in large statistical uncertainty and do not change even we set the skills learnt to 0 on half of the tasks. A better alternative would be using *interquartile mean* in (c).

 -  Rather than number of games (which might be highly variable on random seeds used) in (d), a reasonable alternative could be the *average probability of improvement* over the baseline method -- this will capture how likely the presented method outperforms existing method averaged across all tasks and runs. This also needs to be reported with confidence intervals as it would change depending on the random seeds used.

 - The plot in (b) needs to have error bars (what you are reporting is comparison of two random variables without any uncertainty estimates). Same is true for for plot in (e).

[1] Agarwal, R., Schwarzer, M., Castro, P.S., Courville, A. and Bellemare, M.G., 2021. Deep reinforcement learning at the edge of the statistical precipice. In NeurIPS. https://openreview.net/forum?id=uqv8-U4lKBe

---

> ### Author Response · Authors · 2021-11-19
> **Thanks for the feedback!**
>
> Thank you for taking the time to read our paper and provide such well thought out feedback.
>
> We generally agree with your suggestions and have updated the paper to reflect that. Specifically, we have:
> 1. Replaced the original Fig 4a and 4b with performance profiles for the number of skills learnt and the relative skill boost, respectively (now Fig 4 center and right), including stratified bootstrap confidence intervals (CIs).
> 2. Replaced mean with interquartile mean and added stratified bootstrap CIs to the learning curves in original Fig 4a (now Fig 4 left).
> 3. Replaced the point estimates of reward improvement in original Fig 4d with estimated probabilities of improvement with confidence intervals (now Fig 5 left).
> 4. Replaced the game-by-game point estimates of lifetime coverage improvement in original Fig 4e with performance profiles and stratified bootstrap CIs.
> 5. Moved the skill count and boost histograms (original Fig 4a and 4b) to the appendix (now Fig 7), since robust statistical conclusions there are harder to make. We have nevertheless added error bars to those plots, as requested.
> 6. Cited Agarwal et al 2021 to reflect the source of these recommendations.
>
> Thank you again - we believe that with these changes our analyses better leverage the cross-game structure of our experiments and have resulted in a stronger paper.

---

### Decision · Program_Chairs · 2022-01-20

**Decision:**

Accept (Spotlight)

**Comment:**

This paper tackles the problem of exploration using intrinsic rewards in RL in states that have never been encountered before. The authors derive an information gain auxiliary objective that involves training an ensemble of discriminators and rewarding the policy for their disagreement, which estimates the epistemic uncertainty that comes from the discriminator not having seen enough training examples. The intrinsic reward resulting from the so-called DISDAIN (discriminator disagreement intrinsic reward) exploration bonus is more tailored to the true objective compared to pseudocount-based methods.

Reviewers agree that the paper is well-motivated and well-written, that the proposed DISDAIN exploration method is simple and practical, and that experiments are convincing. Experiments on continuous control tasks such as MuJoCo locomotion environments could have strengthened the paper further.